# Characterization of structure and protein of vitelline membranes of precocial (ring-necked pheasant, gray partridge) and superaltricial (cockatiel parrot, domestic pigeon) birds

Krzysztof Damaziak[1]*, Marek Kieliszek[2], Mateusz Bucław[3]

**1** Department of Animal Breeding, Faculty of Animal Breeding, Bioengineering and Conservation, Institute of Animal Science, University of Life Sciences, Warsaw, Poland, **2** Department of Food Biotechnology and Microbiology, Institute of Food Sciences, Warsaw University of Life Sciences, Warsaw, Poland, **3** Department of Poultry and Ornamental Bird Breeding, Faculty of Biotechnology and Animal Husbandry, West Pomeranian University of Technology Szczecin, Szczecin, Poland

* krzysztof_damaziak@sggw.pl

**Data Availability Statement:** All relevant data are within the paper and its Supporting Information files.

## Abstract

Of all the known oviparous taxa, female birds lay the most diverse types of eggs that differ in terms of shape, shell pigmentation, and shell structure. The pigmentation of the shell, the weight of the egg, and the composition of the yolk correlate with environmental conditions and the needs of the developing embryos. In this study, we analyzed the structure and protein composition of the vitelline membrane (VM) of ring-necked pheasant, gray partridge, cockatiel parrot, and domestic pigeon eggs. We found that the VM structure is characteristic of each species and varies depending on whether the species is precocial (ring-necked pheasant and gray partridge) or superaltrical (cockatiel parrot and domestic pigeon). We hypothesize that a multilayer structure of VM is necessary to counteract the aging process of the egg. The multilayer structure of VM is only found in species with a large number of eggs in one clutch and is characterized by a long incubation period. An interesting discovery of this study is the three-layered VM of pheasant and partridge eggs. This shows that the formation of individual layers of VM in specific sections of the hen's reproductive system is not confirmed in other species. The number of protein fractions varied between 19 and 23, with a molecular weight ranging from 15 to 250 kDa, depending on the species. The number of proteins identified in the VM of the study birds' eggs is as follows: chicken—14, ring-necked pheasant—7, gray partridge—10, cockatiel parrot—6, and domestic pigeon—23. The highest number of species-specific proteins (21) was detected in the VM of domestic pigeon. This study is the first to present the structure and protein composition in the VM of ring-necked pheasant, gray partridge, cockatiel parrot, and domestic pigeon eggs. In addition, we analyzed the relationship between the hatching specification of birds and the structure of the VM.

**Funding:** This research was supported by the Polish project "Proteomic analysis of the vitelline membrane of selected precocial and superartrical avian" (accounting records: 505-10-070300-P00217-99), financed by the Warsaw University of Life Sciences—SGGW, Warsaw, Poland. The funder had no role in study design, data collection and analysis, decision to publish, or preparation of the manuscript.

## Introduction

The vitelline membrane (VM) is a multilayered structure that protects and gives shape to the egg yolk and separates it from the egg white. Together with the chalaza, VM keeps the egg yolk in the central part of the egg, thereby preventing its integration with the shell membranes. In addition, it acts as a diffusion barrier by transporting water and nutrients between the egg yolk and the egg white. It protects the embryo during the first 96 h of incubation against the strongly alkaline nature of the egg white [1, 2].

The specific structure of the VM helps it performs the aforementioned functions. In general, it consists of an inner layer (IL; lamina perivitellina), which is formed before ovulation from the follicular epithelium, and an outer layer (OL; lamina extravitellina), which is formed after ovulation from the mucinous secretion of infundibulum glands (the first segment of the oviduct) [3–6]. The components of IL are expressed by the hepatic cells, as well as granulosa cells, of the female birds. Between the IL and OL lies a granular "continuous membrane" (CM; lamina continua), the composition of which is not known [1]. Electron microscopic results have shown that IL is a single-layered structure and is formed of a network of cylindrical fibers. However, fibrous OL consists of a different number of sublayers [7, 8]. The IL primarily consists of glycoproteins of the zona pellucida, five of which have been identified and described previously [9]. The OL contains numerous proteins analogous to those known as the components of the egg white (ovalbumin, lysozyme C, and ovomucin) and yolk (serum albumin, immunoglobulins, lipovitellin, and apolipoprotein B) [2]. Mann conducted a proteomic analysis and has expanded the number of known VM proteins from 13 [10–12] to 137 [2]. Many of these proteins are VM-specific (ovocalyxin-36, apolipoprotein A-I, ovocleidin-116, semaphorin C3, actin, filamin, clusterin), but their functions remain to be elucidated.

So far, only the data on the structure and protein composition of VM of the hens' eggs are available [1–6]. Much less attention has been paid to the VM of quail eggs [9, 13], and only a few studies have focused on the VM of the eggs of other bird species [7–9]. Chung et al. [7] compared the structure of the VM of hen and duck eggs, and Damaziak et al. [8] compared the structure of the VM of ostrich, emu, and rhea eggs. These authors [7–8] have demonstrated that pattern and thickness of the fibers, as well as the presence of additional structures serving, inter alia, toward the consistency of IL and OL, differ among different species. The differences observed in the strength of VM in different poultry species also suggest different structures that form the membrane [14]. All the previously studied species belong to the precocial group, in which the offsprings are partially independent at clutching, that is, they are covered with down and move and feed themselves but are dependent on parental care. According to our knowledge, there is no data regarding the effect of structure and composition of protein of VM of birds belonging to the other categories on the degree of chick development at the time of hatching. Birds are distinguished into four basic categories: precocial—covered with fluff, seeing, self-feeding, but remaining under the care of parents (e.g. chicken, turkey, duck, and quail); superprecocial—completely independent (e.g. malleefowl and brushturkey); altricial—covered with down and sighted but fed by parents (e.g. seagull); and superaltricial—blind, partially naked, unable to walk in the first few days of life, and completely dependent on parents (e.g. parrot, pigeon, and passerine) [15]. Although this classification was made on the basis of the morphological features and behavior of chicks at clutching, the differences in the structure of the egg and the course of embryogenesis [16–18] themselves suggest differences in the structure of the VM. It is known that eggs from superaltricial birds are characterized by a very small proportion of egg yolk (~20%) to white, unlike eggs from precocial birds, in which case, egg yolks constitute about 45% [19]. Moreover, the eggs of superaltricial birds are incubated immediately after laying, and the entire clutching is characterized by asynchronization, unlike

the eggs of precocial birds, in which case, the incubation begins when the last egg in the clutch is laid. Consequently, most eggs of precocial birds remain in the nest for a few days before the process of incubation begins. It should be mentioned that delayed incubation time has a negative effect on the structure of VM [20].

Therefore, in this study, we hypothesize that the structure of the VM of the eggs of precocial and superaltricial birds is different due to the evolutionary adjustment to clutching behavior. We also identified the proteins present in VM using the NanoAcquity Ultra Performance LC (Waters) system, since the current knowledge on VM proteins is limited. Proteomic analysis might throw some light on the proteins involved in the protection of embryos before and during incubation. The results of this study might be useful to identify birds on par with genetic analysis.

## Materials and methods

### Egg collection

Eggs from three species of precocial birds (ring-necked pheasant (*Phasianus colchicus*), gray partridge (*Perdix perdix*), and laying-type ISA Brown chicken (*Gallus gallus*; only used for proteomic analyses) belonging to Galliformes) and two species of superaltricial birds (cockatiel parrot (*Nymphicus hollandicus*) representing the Psittaciformes and domestic pigeon (*Columba livia*) representing the Columbiformes) were studied. The eggs of pheasant and partridge were obtained from game bird breeding centers (64–061 Kamieniec, PL and 26–070 Łopuszno, PL, respectively), and the eggs of cockatiel parrot and pigeon were obtained from private farms. All birds were kept in breeding flocks, and all eggs were assumed to be fertilized because the breeding flocks included fertile males in them. Each egg was obtained from a different female on the day of laying. A total of 10 eggs from each species of the bird were stored in a refrigerator at 4˚C for 24 h. The egg weight was determined (±0.1 g) after storage, and the samples were taken for analysis. VM was obtained from each egg at 48 h after laying.

### Preparation of the vitelline membrane

For scanning electron microscopy (SEM), VM samples were prepared by following the methodology described by Kirund and McKee [20]. Briefly, six eggs from each of the four species were analyzed yielding a total of 24 samples. The shells of the eggs were broken, and the contents were poured onto a separator to separate the egg yolk from the white. After determining the weight (±0.1 g), the yolk was placed on a glass pan such that the germ disc was visible on the surface. VM was cut with a scalpel around the egg yolk about halfway up. The collected VM was rinsed in deionized water (~4˚C) until all residues of the yolk visible to the naked eye were removed. The weight of the whole VM (±0.1 mg) was determined, and the area of the embryonic disc (not the analyzed area) was separated.

### Scanning electron microscopy

Round fragments of VM (about 2–3 mm in diameter) were cut out for analysis, ensuring that they did not contain any chalaza. The fragments were placed in glass vials containing fixing agent (6 mL of 3% glutaraldehyde solution + 100 mL of 1% paraformaldehyde in 0.1 M potassium phosphate buffer, pH 7.2). Prefixation was performed for 24 h at a storage temperature of 4˚C. Following this, the fragments were fixed in 1% osmium tetroxide ($OsO_4$) prepared in phosphate buffer at room temperature for 1 h. The fixed samples were washed with distilled water and dehydrated by placing it in a series of ethanolic solutions (25, 50, 75, and 95% solutions for one time and 100% solution for three times). The samples were dried with carbon

dioxide and mounted on a stub and coated with 200 Å gold. VM was observed with an FEI QUANTA 200 scanning electron microscope (Hillsboro, OR, USA), operated at 25 kV, at various magnifications.

### Transmission electron microscopy (TEM)

Six eggs from each of the four species were analyzed yielding a total of 24 samples. VM samples were fixed in 2.5% glutaraldehyde for 2 h at 4°C. The fixed samples were washed with phosphate buffer (pH 7.2) for about 2 h at 4°C. Then, the samples were fixed in 2.5% glutaraldehyde for 2 h at 4°C. The fixed samples were washed with phosphate buffer (pH 7.2) for about 2 h at 4°C. Then, the samples were fixed in 1% $OsO_4$ at 4°C for 1 h and dehydrated in an increasing gradient of ethanol and saturated with acetone. Then, the samples were immersed in Epon 812. Following polymerization of the Epon, the samples were cut with a diamond knife on an ultramicrotome (LKB, Sweden) and transferred to copper nets, which were then contrasted in uranyl acetate and lead citrate. The prepared material was examined under TEM (JEM 1220 TEM, JEOL, Japan). From the TEM image, the thickness of the samples was measured and the number of layers of individual bird species was counted using the Nikon optical microscope (type 104c, Japan) equipped with Nis Elements, version 5.10.

### Protein extraction and gel electrophoresis

The obtained VMs were dried in SpeedVac. Eight eggs from each of the 5 species were analyzed yielding a total of 35 samples. Proteins were extracted from the samples by using a buffer consisting of 50 mM Tris–HCl (pH 8.0), 10% glycerol, 2% sodium dodecyl sulfate (SDS), 25 mM ethylenediaminetetraacetic acid, and protease inhibitor cocktail (Sigma-Aldrich, Poland). Samples were incubated overnight under constant stirring and at room temperature. After this, the samples were centrifuged (12000 g, 30 min, 4°C), the supernatant was collected, and the concentration of proteins was determined by using the Lowry method.

Electrophoresis in SDS-polyacrylamide gel (SDS-PAGE) was conducted under denaturing conditions in 4% thickening and 14% separating gels. The samples for analysis were prepared by mixing 15 µL of proteins with 5 µL of reducing buffer and then incubating at 95°C for 5 min under shaking (Eppendorf Thermomixer Comfort, Germany). Electrophoresis was performed in Mini Protean® 3 (Bio-Rad) at a constant current of 20 mA in 1× Tris–glycine buffer (pH 8.3). To visualize protein bands, the gels were stained with Coomassie Brilliant Blue R-250. The electrophoretically separated proteins were documented with the GelDoc 2000 gels registration system (Bio-Rad, France) and analyzed using Quantity One computer program [21, 22].

### Protein identification

Followed by the precipitation of proteins by using acetone, the samples were dissolved in 0.1% RapidGest™ surfactant (Waters) in 50 mM ammonium hydroxide. After the reduction and alkylation of the cysteine residues, the samples were digested at 30°C for 12 h using trypsin (Sigma-Aldrich, Poland). The reaction was stopped by adding trichloroacetic acid at a final concentration of 1% (v/v). Low molecular weight proteins were digested with chymotrypsin (Sigma-Aldrich, Poland) in addition to trypsin. The digested peptides were analyzed using the NanoAcquity Ultra Performance LC (Waters) system combined with a mass spectrometer. Peptides were added to the Symmetry® C18 column (5 µm × 180 µm × 20 mm) (Waters) at a flow rate of 10 µL/min in 99% buffer A (0.1% formic acid in water) and 1% buffer B (0.1% formic acid in acetonitrile) for 3 min. The trapped peptides were separated on the BEH 130 C18 analytical column (1.7 µm × 75 µm × 200 mm) balanced in 97% buffer A and 3% buffer B. The

column was eluted with a linear gradient of buffer B at a constant flow rate of 300 nL/min at 35˚C. Online $MS^E$ analyses were performed in positive ionization mode using the Synapt G2 HDMS mass spectrometer (Waters). The fragmentation spectra were recorded in the range of 50–2000 Da, and the energy of transfer collision was increased in the range of 15–35 V. The accuracy of raw data of molecular weights was corrected with leucine enkephalin (flow rate 2 ng/μL, 1 μL/min, 556.2771 Da/e [M+H]+). Each sample was analyzed at least thrice and mixed with bovine albumin (60 fmol) as an internal standard during the tryptic digestion process. To identify the proteins, peak lists were created from raw datasets and used to search for proteins in a database using Protein Lynx Global Server, version 2.4 (Waters) [23].

## Statistical analysis

All the analyzed traits were compared between the species using Duncan's test at a significance level of $P \leq 0.05$. Calculations were performed using Statistica 12 software [24].

## Results

### Characteristics of egg and VM

Table 1 (S1 Data) presents a comparison of the weight of the eggs between the examined species of precocial and superaltricial birds. The weight of the egg and yolk, the proportion of yolk weight to the egg weight, and the weight of the VM were found to be significantly higher in precocial birds than that of superaltricial birds ($P < 0.05$). Among the precocial birds, egg weight and yolk weight were higher in ring-necked pheasant eggs than that of the gray partridge eggs ($P < 0.05$). However, gray partridge eggs had a much higher content of egg yolk than that of ring-necked pheasant eggs ($P = 0.025$). Among the superaltricial birds, egg weight, yolk weight, and VM weight were found to be higher in pigeon eggs than that of cockatiel parrot eggs ($P < 0.05$).

The highest proportion of the weight of VM in the weight of egg yolk was found to be in pigeon eggs, followed by the ring-necked pheasant and cockatiel parrot eggs, and the lowest proportion was found in gray partridge eggs ($P < 0.05$). The VM in the egg yolk of precocial birds was significantly thicker than that of egg yolk of superaltricial birds ($P < 0.05$). A thicker VM was observed in the egg yolks of ring-necked pheasant eggs than that of egg yolks of gray partridge ($P = 0.016$), and the VM in the egg yolk of pigeon eggs was thicker than that of egg yolk of cockatiel parrot eggs ($P = 0.001$).

**Table 1. Results (mean ± SD) of the comparative analysis of the egg and yolk weights and VM characteristics of eggs from some precocial and superaltricial birds.**

| Items | | Species | | | |
|---|---|---|---|---|---|
| | | Ring-necked pheasant | Gray partridge | Cockatiel parrot | Domestic pigeon |
| Egg weight | (g) | 31.82 ± 0.59[d] | 19.53 ± 1.67[c] | 4.70 ± 0.26[a] | 16.70 ± 0.81[b] |
| Yolk weight | (g) | 10.30 ± 0.20[d] | 7.46 ± 0.47[c] | 0.97 ± 0.06[a] | 3.33 ± 0.13[b] |
| Yolk ratio[1] | (%) | 32.37 ± 0.56[b] | 38.26 ± 1.21[c] | 20.61 ± 0.64[a] | 19.98 ± 1.10[a] |
| VM weight | (g) | 6.70 ± 0.46[c] | 6.50 ± 0.42[c] | 3.89 ± 0.26[b] | 4.61 ± 0.33[a] |
| VM ratio[2] | (%) | 20.69 ± 1.45[c] | 17.03 ± 1.48[a] | 18.92 ± 1.52[b] | 23.09 ± 1.62[d] |
| VM thickness | (μm) | 37.68 ± 0.45[d] | 33.59 ± 0.77[c] | 7.33 ± 0.56[a] | 21.75 ± 1.14[b] |

[a–d]Means within a row without a common superscript differ significantly, $P < 0.05$

VM = vitelline membrane

SD = standard deviation

[1]Yolk weight ratio to egg weight

[2]VM weight ratio to yolk weight

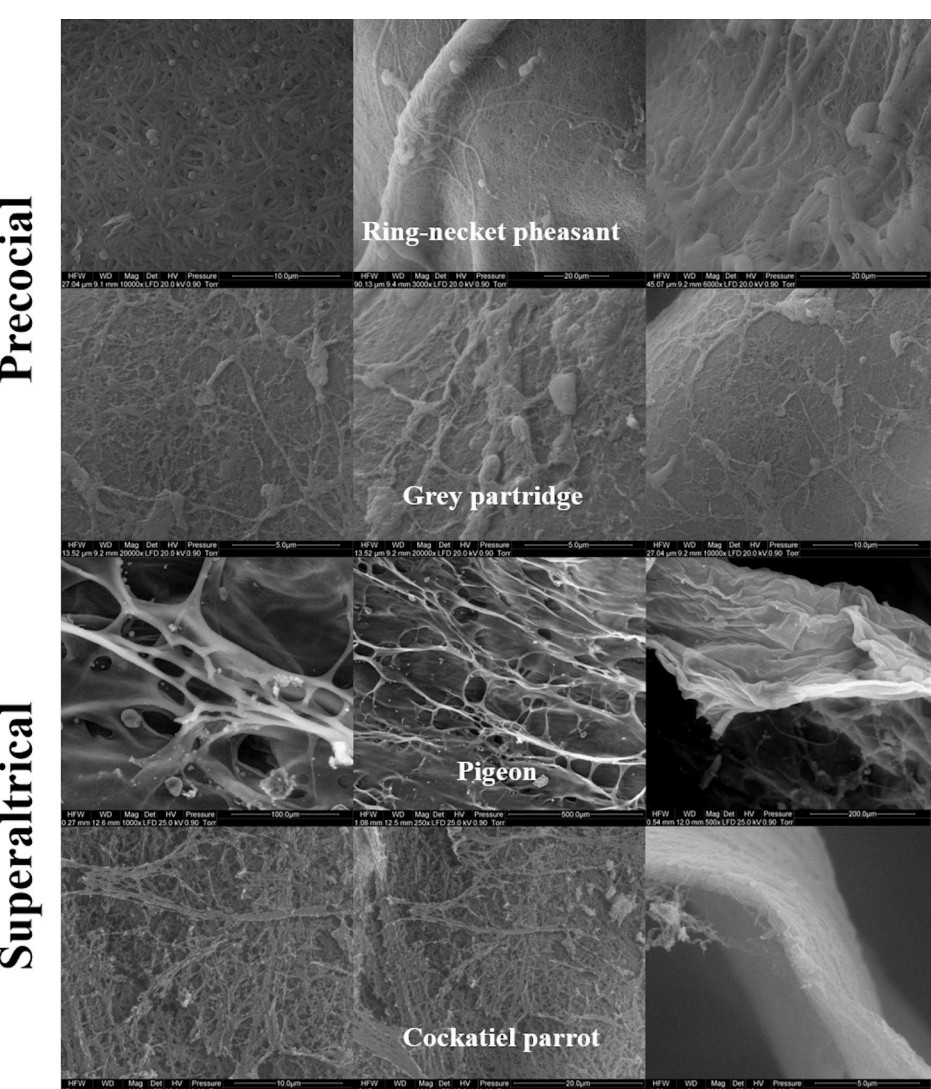

**Fig 1. Scanning electron micrograph.** Outer layers of the vitelline membrane in the egg yolk of precocial (ring-necked pheasant and gray partridge) and superaltricial (pigeon and cockatiel parrot) birds.

## VM structure

Figs 1 and 2 show the SEM images of the structure of the VM of egg yolks of the studied bird species. The structure of the OL (Fig 1) of ring-necked pheasant and gray partridge eggs was found to be uniformly formed by thin and thick fibers of protein that were densely arranged. The course of the fibers formed a three-dimensional network along the lines of a truss. A similar structure was observed for the OL of cockatiel parrot eggs, but the fibers showed a uniform thickness (Fig 1). A completely different structure of OL was observed in the case of pigeon eggs, as the OL in this species did not have a fibrous structure and was entirely formed from strongly branched sheets. The branches of the sheets were not regular and had a few pores of a much larger diameter than that of the pores in the networks of OL fiber of other examined bird species. However, when observed from the inside, IL did not show a typical fibrous structure in any of the examined species, even at a magnification of up to ×10000 under the SEM (Fig 2). In the case of ring-necked pheasant, gray partridge, and pigeon eggs, the IL was similar

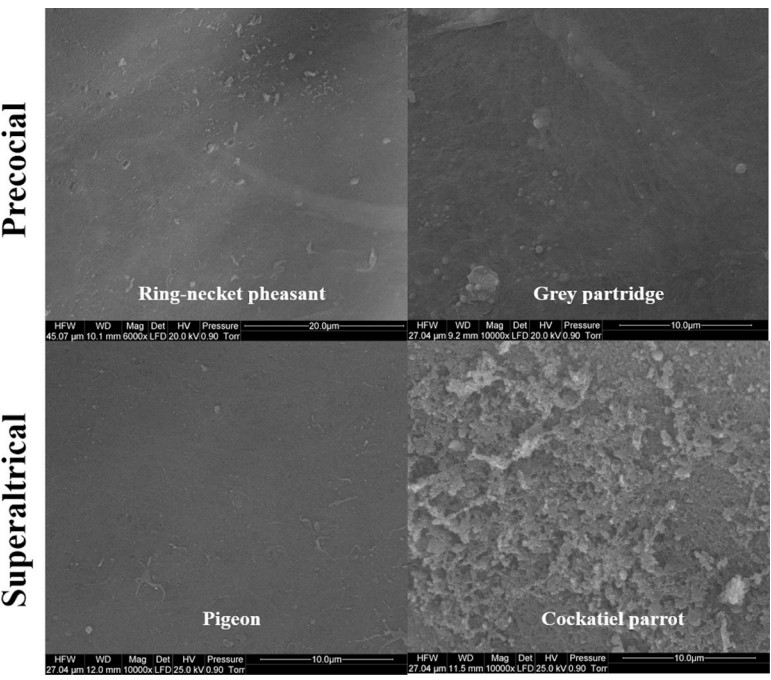

**Fig 2. Scanning electron micrograph.** Inner layers of the vitelline membrane in the egg yolk of precocial (ring-necked pheasant and gray partridge) and superaltricial (pigeon and cockatiel parrot) birds.

and appeared like a homogeneous layer of the membrane. In contrast, the IL of the cockatiel parrot eggs was made up of densely arranged protein grains with an irregular structure (Fig 2).

In the TEM image, the structure of the VM of ring-necked pheasant and gray partridge eggs showed an analogous three-layered structure (Fig 3). In both species, it was possible to distinguish the three primary layers of VM formed by IL ($IL_{1-3}$) and OL ($OL_{1-3}$). It was also possible to distinguish a few sublayers of different thicknesses in the cross-section of the main VM layers. The difference in the VM structure between ring-necked pheasant and gray partridge eggs was visible during the course and continuity of IL and OL. In the VM of ring-necked pheasant eggs, both $IL_{1-3}$ and $OL_{1-3}$ ran strictly parallel, whereas in the VM of gray partridge eggs, numerous branches of individual layers and blindly ended deviations giving an impression of internal connectors were observed in the cross-section (Fig 4). The cross-section of the whole width of the VM of cockatiel parrot eggs formed a single layer as observed in the case of ring-necked pheasant and gray partridge eggs. The TEM image of the cross-section of the VM of pigeon eggs indicated a completely different structure, in which case, the OL and IL were conventionally distinguished, but their cross-section differed significantly from the cross-section of the VM of other discussed species (Fig 3). In general, the VM of the pigeon eggs had 15–18 sublayers forming the OL and a similar number of sublayers, but with a twofold greater thickness, forming the IL. However, all the layers were much loose than that of the closely adjacent layers observed in the VM of the other examined bird species. Since the OL of pigeon VM was formed solely from sheets and not from longitudinal fibers, as in the case of ring-necked pheasant, gray partridge, and cockatiel parrot eggs, it appeared as a more homogeneous structure with less porosity.

## VM proteome

Fig 5 shows the electrophoretic separation of the proteins isolated from VM. The selected protein bands (red arrows) were marked, for the additional identification of the proteins.

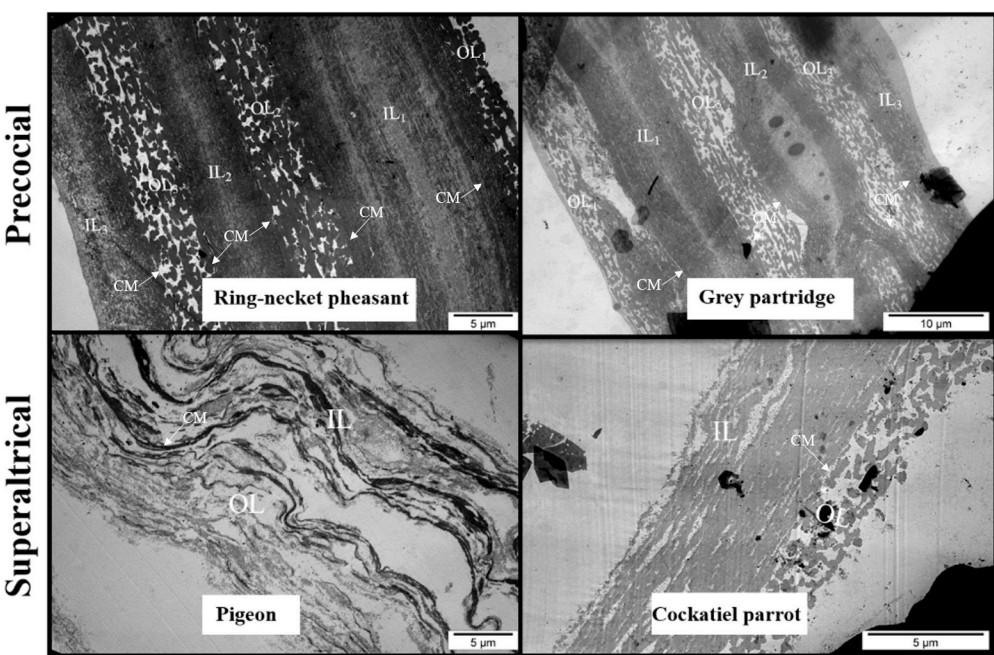

**Fig 3. Transmission electron micrograph.** The vitelline membrane of the egg yolk of precocial (ring-necked pheasant and gray partridge) and superaltricial (pigeon and cockatiel parrot) birds. OL = outer layer; CM = continuous membrane; IL = inner layer.

Significant differences were found between the protein bands after electrophoretic separation. In the first lane that had electrophoretically separated VM of pigeon egg yolks, a total of about 23 protein bands was observed. In the case of the VM of cockatiel parrot, gray partridge, and ring-necked pheasant eggs, the number of protein bands observed was 20, 19, and 22, respectively. The highest variability between the proteins of the VM of the birds was found for the fractions with molecular weights ranging from 37 to 100 kDa. Moreover, the protein bands (37–15 kDa) obtained from the VM of parrot and partridge eggs were found to be of less intensity compared to the VM of other bird species. A similar intensity of about 46 kDa was observed for the protein bands of pigeon and cockatiel parrot eggs. The VM protein bands of partridge and pheasant eggs in the range of high molecular weight were very similar, and the differences were observed only in the case of low molecular weight proteins. In pheasant eggs,

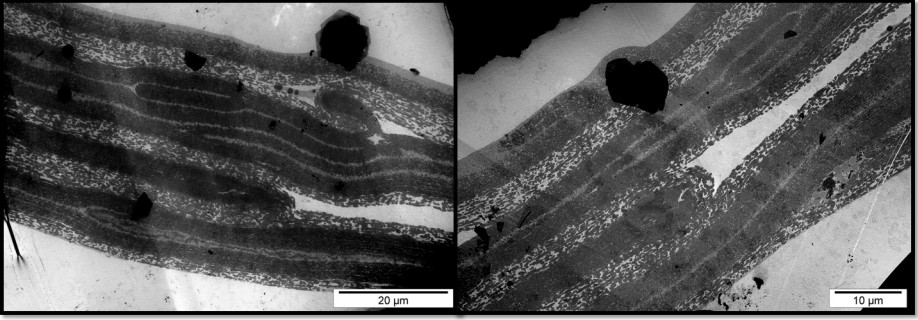

**Fig 4. Transmission electron micrograph.** Cross-section of the vitelline membrane of the egg yolk of gray partridge eggs.

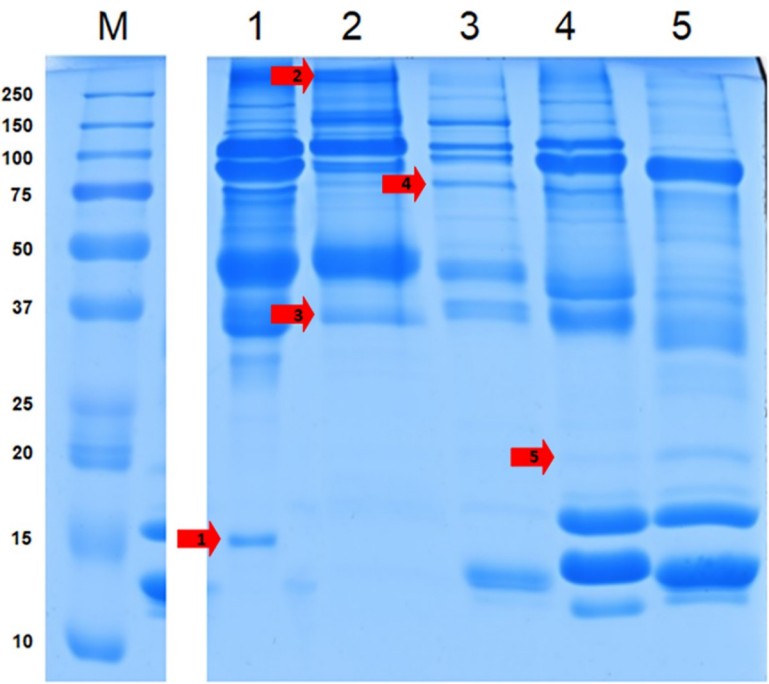

**Fig 5. Sodium dodecyl sulfate-polyacrylamide gel electrophoresis of whole vitelline membrane.** Analysis of proteins of the vitelline membrane in the egg yolk of superaltricial (1: pigeon and 2: cockatiel parrot) and precocial (3: gray partridge, 4: ring-necked pheasant, and 5: chicken) birds. The red arrows indicate the protein bands selected for detailed analysis.

bands of 17, 20, and 23 kDa were observed, which were not found in partridge. It should be noted that the pheasant VM resulted in a protein band of 11 kDa, which was not found in other birds (excluding hen). Similar protein profiles were found for the VM of hen and pheasant eggs in the low molecular weight range (20–10 kDa), which may indicate the similarity between the two species. Thus, the obtained results showed that the number of protein bands obtained from VM on electropherograms depended on the species of the birds.

Tables 2 and 3 and (S2 Data and S3 Data) show the results of the proteomic identification of VM protein fractions of the studied birds.

The analysis of the characteristics of the proteins in the VM of hen eggs using a Venn diagram (Fig 6) revealed 14 proteins, which were not identified in the VM of other birds. In the VM of ring-necked pheasant eggs, seven proteins were identified, which were absent in the eggs of other birds. It should be emphasized that the protein structure of the VM pf pheasant eggs was more closely related to other birds in terms of the proteins present in the VM as well. The most pronounced similarity with the VM of pheasant eggs was found for gray partridge (five proteins) and hen eggs (two proteins). Among the studied avian species, the lowest number of proteins (six) was found in the VM of cockatiel parrot eggs. It should also be emphasized that one protein (A0A2IOTKM1) was found commonly in cockatiel parrot, ring-necked pheasant, and pigeon eggs. The VM of pigeon eggs demonstrated the highest number of protein bands (23). Proteomic analysis of the protein fractions of about 170 and 15 kDa obtained from the VM of pigeon eggs showed the presence of zona pellucida sperm-binding protein 1 (ZP1) and keratin protein (type II cytoskeletal 5-like) in the keratin structure of the VM, respectively.

In addition, individual protein bands obtained through electrophoretic separation (Fig 5, marked with red arrows) were selected and subjected to an in-depth analysis to detect the

**Table 2. The proteomic analysis of water-washed vitelline membrane (VM) of selected bird species.** All proteins in the whole VMs were identified by the sodium dodecyl sulfate-polyacrylamide gel electrophoresis (SDS-PAGE).

| Swiss-Prot/ Trembl accession | Protein | mW (Da) | pI (pH) | PLGS score | Peptides | Theoretical peptides | Coverage (%) | Products | Digest peptides | Protein ID |
|---|---|---|---|---|---|---|---|---|---|---|
| **PRECOCIAL** | | | | | | | | | | |
| **Chicken (layer hen); *Gallus gallus*** | | | | | | | | | | |
| P00698 | LYSC CHICK Lysozyme C OS *Gallus gallus* OX 9031 GN LYZ PE 1 SV 1 | 16228 | 9.2 | 18802.8 | 7 | 12 | 36.1 | 252 | 7 | 5495 |
| tr | A0A2P4SB66 BAMTH Uncharacterized protein OS *Bambusicola thoracicus* OX 9083 GN CIB84 01490 | 20279 | 8.5 | 12399.5 | 14 | 16 | 47.0 | 250 | 11 | 789063 |
| tr | A0A140JXP0 CHICK Zona pellucida sperm-binding protein 1 OS *Gallus gallus* OX 9031 GN ZP1 PE | 102171 | 8.3 | 9989.5 | 12 | 27 | 11.0 | 226 | 9 | 290653 |
| tr | A0A2H4Y814 CHICK OVA Fragment OS *Gallus gallus* OX 9031 GN OVA PE 2 SV 1 | 42838 | 4.9 | 9123.7 | 9 | 27 | 38.6 | 185 | 9 | 282735 |
| P01012 | OVAL CHICK Ovalbumin OS *Gallus gallus* OX 9031 GN SERPINB14 PE 1 SV 2 | 42853 | 5.0 | 8592.4 | 9 | 27 | 38.6 | 186 | 9 | 3671 |
| P79762 | ZP3 CHICK Zona pellucida sperm-binding protein 3 OS *Gallus gallus* OX 9031 GN ZP3 PE 1 SV 4 | 46736 | 5.9 | 7070.4 | 18 | 22 | 23.1 | 225 | 12 | 9092 |
| tr | A0A1D5P1X2 CHICK Clusterin OS *Gallus gallus* OX 9031 GN CLU PE 3 SV 1 | 53785 | 5.4 | 3729.2 | 15 | 38 | 30.5 | 139 | 14 | 319290 |
| tr | A0A146J2U8 CHICK Protein TENP OS *Gallus gallus* OX 9031 GN TENP PE 2 SV 1 | 47387 | 5.6 | 2215.3 | 4 | 24 | 11.2 | 60 | 4 | 400552 |
| P53478 | ACT5 CHICK Actin cytoplasmic type 5 OS *Gallus gallus* OX 9031 PE 3 SV 1 | 41808 | 5.1 | 1481.4 | 4 | 34 | 15.4 | 31 | 4 | 7156 |
| tr | A0A0K0PUH6 CHICK Chemerin OS *Gallus gallus* OX 9031 GN RARRES2 PE 2 SV 1 | 18219 | 9.0 | 832.9 | 5 | 17 | 27. 8 | 35 | 5 | 344118 |
| Q25C36 | OLFL3 CHICK Olfactomedin like protein 3 OS *Gallus gallus* OX 9031 GN OLFML3 PE 2 SV 1 | 44817 | 5.7 | 404.5 | 5 | 31 | 11.2 | 34 | 5 | 3280 |
| P01875 | IGHM CHICK Ig mu chain C region OS *Gallus gallus* OX 9031 PE 2 SV 2 | 48142 | 6.0 | 290.5 | 3 | 26 | 11.2 | 15 | 3 | 5166 |
| tr | A0A140T8F5 CHICK Polymeric immunoglobulin receptor OS *Gallus gallus* OX 9031 GN PIGR PE 4 | 70526 | 4.8 | 260.9 | 3 | 42 | 7.9 | 25 | 3 | 145853 |
| P02845 | VIT2 CHICK Vitellogenin 2 OS *Gallus gallus* OX 9031 GN VTG2 PE 1 SV 1 | 204677 | 9.3 | 174.8 | 8 | 131 | 5.8 | 42 | 8 | 1087 |
| tr | A0A087VPD3 BALRE Vitellogenin 2 Fragment OS *Balearica regulorum gibbericeps* OX 100784 G | 201826 | 9.2 | 138.2 | 5 | 123 | 3.6 | 30 | 5 | 306649 |
| P02789 | TRFE CHICK Ovotransferrin OS *Gallus gallus* OX 9031 PE 1 SV 2 | 77726 | 6.8 | 133.3 | 2 | 74 | 4.1 | 13 | 2 | 1940 |
| Q98UI9 | MUC5B CHICK Mucin 5B OS *Gallus gallus* OX 9031 GN MUC5B PE 1 SV 1 | 233393 | 5.2 | 56.8 | 2 | 139 | 1.3 | 22 | 2 | 2906 |
| tr | A0A1D5P2X2 CHICK Alpha-2-macroglobulin-like 1 OS *Gallus gallus* OX 9031 PE 4 SV 1 | 163357 | 8.0 | 36.1 | 2 | 99 | 1.6 | 16 | 2 | 145034 |
| **Ring-necked pheasant; *Phasianus colchicus*** | | | | | | | | | | |
| P00702 | LYSC PHACO Lysozyme C OS *Phasianus colchicus* colchicus OX 9057 GN LYZ PE 1 SV 2 | 16154 | 9.2 | 6282.7 | 7 | 13 | 40.8 | 121 | 7 | 5343 |
| tr | Q4VTT5 PHACC Zona pellucida c OS *Phasianus colchicus* OX 9054 PE 4 SV 1 | 47813 | 5.2 | 5835.7 | 11 | 24 | 21.1 | 142 | 9 | 357975 |

*(Continued)*

**Table 2.** (Continued)

| Swiss-Prot/ Trembl accession | Protein | mW (Da) | pI (pH) | PLGS score | Peptides | Theoretical peptides | Coverage (%) | Products | Digest peptides | Protein ID |
|---|---|---|---|---|---|---|---|---|---|---|
| tr | A0A140JXP0 CHICK Zona pellucida sperm-binding protein 1 OS *Gallus gallus* OX 9031 GN ZP1 PE | 102171 | 8.3 | 5648.6 | 9 | 27 | 9.1 | 149 | 7 | 290653 |
| P79762 | ZP3 CHICK Zona pellucida sperm-binding protein 3 OS *Gallus gallus* OX 9031 GN ZP3 PE 1 SV 4 | 46736 | 5.9 | 5294.3 | 10 | 22 | 18.1 | 126 | 8 | 9092 |
| tr | A0A091UKP2 PHORB Vitelline membrane outer layer protein 1 OS *Phoenicopterus ruber* ruber O | 19988 | 6.8 | 4644.9 | 9 | 12 | 30.1 | 116 | 6 | 608332 |
| P41366 | VMO1 CHICK Vitelline membrane outer layer protein 1 OS *Gallus gallus* OX 9031 GN VMO1 PE 1 SV 1 | 20221 | 8.5 | 4493.6 | 13 | 16 | 41.0 | 169 | 10 | 9423 |
| tr | G1NME9 MELGA Alpha-2-macroglobulin-like 1 OS *Meleagris gallopavo* OX 9103 GN A2ML1 PE 4 SV 2 | 163475 | 8.9 | 2235.6 | 19 | 95 | 12.1 | 234 | 17 | 508606 |
| tr | A0A2H4Y7W8 CHICK OVA Fragment OS *Gallus gallus* OX 9031 GN OVA PE 2 SV 1 | 42680 | 5.4 | 2178.5 | 1 | 27 | 5.2 | 37 | 1 | 250032 |
| tr | A0A226MZF6 CALSU Clusterin OS *Callipepla squamata* OX 9009 GN ASZ78 001440 PE 3 SV 1 | 51670 | 5.4 | 2127.0 | 9 | 40 | 21.8 | 83 | 9 | 489139 |
| P53478 | ACT5 CHICK Actin cytoplasmic type 5 OS *Gallus gallus* OX 9031 PE 3 SV 1 | 41808 | 5.1 | 1991.00 | 7 | 34 | 20.2 | 55 | 6 | 7156 |
| tr | A0A226N891 CALSU Uncharacterized protein OS *Callipepla squamata* OX 9009 GN ASZ78 008997 P | 84169 | 5.4 | 1449.8 | 8 | 66 | 8.0 | 96 | 7 | 478906 |
| tr | G1MYK6 MELGA Ovalbumin OS *Meleagris gallopavo* OX 9103 GN SERPINB14 PE 3 SV 2 | 42988 | 5.0 | 1060.2 | 2 | 26 | 8.5 | 43 | 2 | 502906 |
| tr | G1MVV5 MELGA Ovotransferrin OS *Meleagris gallopavo* OX 9103 GN TF PE 3 SV 2 | 77727 | 6.6 | 723.6 | 6 | 72 | 11.5 | 47 | 6 | 498065 |
| Q98UI9 | MUC5B CHICK Mucin 5B OS *Gallus gallus* OX 9031 GN MUC5B PE 1 SV 1 | 233393 | 5.2 | 310.7 | 3 | 139 | 2.2 | 41 | 3 | 2906 |
| tr | A0A1V4JWT9 PATFA Ovalbumin-related protein X OS *Patagioenas fasciata* monilis OX 372326 GN | 44274 | 7.8 | 271.7 | 5 | 30 | 3.1 | 27 | 1 | 130111 |
| tr | A0A091G4Q7 9AVES Alpha-2-macroglobulin-like 1 Fragment OS *Cuculus canorus* OX 55661 GN N | 107871 | 6.5 | 267.4 | 2 | 61 | 2.5 | 26 | 2 | 173711 |
| tr | A0A2I0TKM1 LIMLA Type II cytoskeletal 5-like OS *Limosa lapponica baueri* OX 1758121 GN lla | 63071 | 5.0 | 231.8 | 2 | 55 | 2. 5 | 50 | 2 | 528417 |
| P01013 | OVALX CHICK Ovalbumin-related protein X Fragment OS *Gallus gallus* OX 9031 GN SERPINB14C PE 3 SV 1 | 26274 | 4.9 | 185.6 | 1 | 15 | 5.2 | 11 | 1 | 3669 |
| **Gray partridge;** *Pedrix pedrix* | | | | | | | | | | |
| tr | G1MYK6 MELGA Ovalbumin OS *Meleagris gallopavo* OX 9103 GN SERPINB14 PE 3 SV 2 | 42988 | 5.0 | 7342.1 | 7 | 26 | 31.3 | 122 | 7 | 502906 |
| tr | G1NMV6 MELGA Clusterin OS *Meleagris gallopavo* OX 9103 GN CLU PE 3 SV 2 | 35883 | 6.6 | 6536.1 | 11 | 28 | 29.6 | 143 | 9 | 552529 |
| tr | A0A2P4SV45 BAMTH Uncharacterized protein Fragment OS *Bambusicola thoracicus* OX 9083 GN | 46710 | 5.4 | 6413.0 | 2 | 27 | 8.6 | 59 | 2 | 747230 |

(*Continued*)

**Table 2.** (*Continued*)

| Swiss-Prot/ Trembl accession | Protein | mW (Da) | pI (pH) | PLGS score | Peptides | Theoretical peptides | Coverage (%) | Products | Digest peptides | Protein ID |
|---|---|---|---|---|---|---|---|---|---|---|
| tr | P84496 ALOAE Lysozyme OS *Alopochen aegyptiaca* OX 30382 PE 3 SV 1 | 14408 | 9.5 | 5976.3 | 8 | 11 | 25.6 | 117 | 4 | 689231 |
| tr | A5HTY5 COTCO ZP1 protein Fragment OS *Coturnix coturnix* OX 9091 GN ZP1 PE 2 SV 1 | 101001 | 8.0 | 5353.4 | 9 | 30 | 8.2 | 150 | 6 | 33861 |
| tr | Q4VTT5 PHACC Zona pellucida c OS *Phasianus colchicus* OX 9054 PE 4 SV 1 | 47813 | 5.2 | 5352.3 | 12 | 24 | 21.1 | 169 | 9 | 357975 |
| tr | Q4VTT4 9GALL Zona pellucida c OS *Lyrurus tetrix* OX 1233216 PE 4 SV 1 | 47747 | 5.7 | 5170.9 | 12 | 24 | 22.6 | 155 | 10 | 201628 |
| tr | G1MVV5 MELGA Ovotransferrin OS *Meleagris gallopavo* OX 9103 GN TF PE 3 SV 2 | 77727 | 6.6 | 3204.9 | 15 | 72 | 20.4 | 155 | 14 | 498065 |
| O42273 | TENP CHICK Protein TENP OS *Gallus gallus* OX 9031 GN TENP PE 2 SV 1 | 47404 | 5.5 | 3146.7 | 1 | 25 | 4.8 | 39 | 1 | 9683 |
| tr | G1NME9 MELGA Alpha-2-macroglobulin-like 1 OS *Meleagris gallopavo* OX 9103 GN A2ML1 PE 4 SV 2 | 163475 | 8.9 | 2869.8 | 21 | 95 | 16.6 | 263 | 20 | 508606 |
| tr | A0A2P4TFF1 BAMTH Uncharacterized protein OS *Bambusicola thoracicus* OX 9083 GN CIB84 00115 | 185200 | 8.0 | 1146.6 | 7 | 111 | 4.5 | 82 | 6 | 740122 |
| tr | A0A226MNG6 CALSU Uncharacterized protein OS *Callipepla squamata* OX 9009 GN ASZ78 006884 P | 49818 | 4.9 | 962.9 | 2 | 41 | 5.9 | 22 | 2 | 510342 |
| tr | A0A087R1Y4 APTFO Uncharacterized protein Fragment OS *Aptenodytes forsteri* OX 9233 GN AS | 26013 | 4.7 | 790.6 | 4 | 14 | 5.2 | 20 | 1 | 341341 |
| tr | A0A2I0LY02 COLLI Alpha-2-macroglobulin-like protein 1 OS *Columba livia* OX 8932 GN A306 00 | 165215 | 9.2 | 743.6 | 4 | 95 | 3.0 | 52 | 4 | 59852 |
| P53478 | ACT5 CHICK Actin cytoplasmic type 5 OS *Gallus gallus* OX 9031 PE 3 SV 1 | 41808 | 5.1 | 464.7 | 3 | 34 | 12.0 | 21 | 3 | 7156 |
| Q98UI9 | MUC5B CHICK Mucin 5B OS *Gallus gallus* OX 9031 GN MUC5B PE 1 SV 1 | 233393 | 5.2 | 126.7 | 3 | 139 | 1.9 | 30 | 3 | 2906 |
| tr | A0A091G4Q7 9AVES Alpha-2-macroglobulin-like 1 Fragment OS *Cuculus canorus* OX 55661 GN N | 107871 | 6.5 | 104.2 | 2 | 61 | 2.5 | 21 | 2 | 173711 |
| tr | G3UU60 MELGA Uncharacterized protein OS *Meleagris gallopavo* OX 9103 PE 4 SV 1 | 75040 | 8.7 | 62.9 | 3 | 59 | 6.0 | 15 | 3 | 846630 |
| **SUPERALTRICIAL** | | | | | | | | | | |
| **Domestic pigeon; *Columba liva*** | | | | | | | | | | |
| tr | A0A2I0MW20 COLLI Ovalbumin-related protein X OS *Columba livia* OX 8932 GN A306 00001787 PE | 43832 | 7.9 | 15955.0 | 39 | 31 | 61.3 | 546 | 25 | 59432 |
| tr | A0A2I0LY02 COLLI Alpha-2-macroglobulin-like protein 1 OS *Columba livia* OX 8932 GN A306 00 | 165215 | 9.2 | 12830.3 | 59 | 95 | 31.9 | 1050 | 48 | 59852 |
| tr | A0A2I0M6I1 COLLI Zona pellucida sperm-binding protein 3 OS *Columba livia* OX 8932 GN A306 00 | 47079 | 6.5 | 8856.7 | 12 | 29 | 15.2 | 168 | 7 | 51054 |
| tr | A0A1V4JAY6 PATFA Uncharacterized protein OS *Patagioenas fasciata monilis* OX 372326 GN AV5 | 119578 | 9.0 | 7324.3 | 28 | 60 | 20.6 | 464 | 22 | 631533 |

(*Continued*)

**Table 2.** (*Continued*)

| Swiss-Prot/ Trembl accession | Protein | mW (Da) | pI (pH) | PLGS score | Peptides | Theoretical peptides | Coverage (%) | Products | Digest peptides | Protein ID |
|---|---|---|---|---|---|---|---|---|---|---|
| tr | A0A1V4JYZ2 PATFA Vitellogenin 2-like OS *Patagioenas fasciata monilis* OX 372326 GN AV530 0 | 175956 | 8.2 | 5428.3 | 21 | 106 | 15.7 | 254 | 20 | 288584 |
| tr | A0A2I0LGQ9 COLLI Uncharacterized protein OS *Columba livia* OX 8932 GN A306 00000120 PE 4 S | 34862 | 5.5 | 5357.3 | 10 | 18 | 21.0 | 153 | 7 | 59471 |
| tr | R7VU42 COLLI Ig gamma chain C region Fragment OS *Columba livia* OX 8932 GN A306 11075 PE 4 S | 10819 | 4.5 | 4089.1 | 4 | 8 | 34.0 | 61 | 3 | 56564 |
| tr | A0A094KPL4 ANTCR Alpha-2-macroglobulin-like 1 Fragment OS *Antrostomus carolinensis* OX 2 | 80882 | 9.6 | 3225.3 | 8 | 49 | 7.7 | 134 | 5 | 737869 |
| tr | A0A2I0LIU5 COLLI Lysozyme OS *Columba livia* OX 8932 GN LYZ PE 3 SV 1 | 16950 | 9.7 | 1297.8 | 5 | 15 | 23.3 | 45 | 4 | 50139 |
| tr | A0A2I0MDV9 COLLI Vitellogenin 1 OS *Columba livia* OX 8932 GN A306 00005842 PE 4 SV 1 | 155410 | 9.9 | 1280.7 | 25 | 104 | 24.8 | 271 | 24 | 60028 |
| tr | U3JPW5 FICAL Uncharacterized protein OS *Ficedula albicollis* OX 59894 PE 3 SV 1 | 59786 | 5.6 | 1149.3 | 5 | 41 | 10.3 | 51 | 5 | 88292 |
| tr | A0A1V4JZ59 PATFA Vitellogenin 1 OS *Patagioenas fasciata monilis* OX 372326 GN AV530 007742 | 188213 | 8.6 | 1140.6 | 26 | 132 | 18.0 | 237 | 25 | 130590 |
| tr | H0Z0C5 TAEGU Uncharacterized protein OS *Taeniopygia guttata* OX 59729 PE 3 SV 1 | 55310 | 4.8 | 1081.6 | 3 | 37 | 5.4 | 36 | 3 | 456330 |
| P63256 | ACTG ANSAN Actin cytoplasmic 2 OS *Anser anser anser* OX 8844 GN ACTG1 PE 2 SV 1 | 41850 | 5.2 | 852.9 | 9 | 34 | 34.7 | 49 | 8 | 7055 |
| tr | A0A2I0MWA2 COLLI Ovalbumin-like OS *Columba livia* OX 8932 GN A306 00001789 PE 3 SV 1 | 42895 | 4.6 | 832.3 | 9 | 25 | 38.6 | 64 | 8 | 62894 |
| tr | A0A2I0LJ29 COLLI Uncharacterized protein Fragment OS *Columba livia* OX 8932 GN A306 0000 | 24662 | 6.6 | 504.8 | 2 | 19 | 11.4 | 16 | 2 | 687600 |
| tr | A0A091VIN1 NIPNI Alpha tectorin Fragment OS *Nipponia nippon* OX 128390 GN Y956 09212 PE | 46906 | 7.1 | 461.7 | 5 | 27 | 7.9 | 29 | 4 | 182954 |
| tr | A0A1V4JQ70 PATFA Uncharacterized protein OS *Patagioenas fasciata monilis* OX 372326 GN AV5 | 18384 | 8.4 | 451.0 | 2 | 15 | 18.1 | 15 | 2 | 636963 |
| tr | A0A2I0LN01 COLLI BPI fold-containing family B member 2 OS *Columba livia* OX 8932 GN BPIFB | 54428 | 4.6 | 352.7 | 4 | 23 | 11.2 | 30 | 4 | 686748 |
| tr | A0A2I0LY07 COLLI Alpha-2-macroglobulin-like protein 1 OS *Columba livia* OX 8932 GN A306 00 | 145725 | 8.0 | 340.1 | 9 | 86 | 9.8 | 67 | 9 | 692380 |
| tr | A0A2I0TKM1 LIMLA Type II cytoskeletal 5-like OS *Limosa lapponica baueri* OX 1758121 GN lla | 63071 | 5.0 | 299.9 | 3 | 55 | 3.7 | 36 | 3 | 528417 |
| tr | R7VRC4 COLLI Complement C3 OS *Columba livia* OX 8932 GN A306 14901 PE 4 SV 1 | 181394 | 6.6 | 271.1 | 5 | 134 | 4.5 | 36 | 4 | 55053 |

(*Continued*)

**Table 2.** (Continued)

| Swiss-Prot/ Trembl accession | Protein | mW (Da) | pI (pH) | PLGS score | Peptides | Theoretical peptides | Coverage (%) | Products | Digest peptides | Protein ID |
|---|---|---|---|---|---|---|---|---|---|---|
| tr | A0A2I0LIU4 COLLI Uncharacterized protein Fragment OS *Columba livia* OX 8932 GN A306 0000 | 51191 | 5.9 | 267.9 | 4 | 26 | 12.9 | 27 | 4 | 62521 |
| tr | A0A1V4KQV2 PATFA Ovotransferrin OS *Patagioenas fasciata monilis* OX 372326 GN TF PE 3 SV 1 | 77253 | 6.9 | 194.4 | 6 | 68 | 12.5 | 42 | 6 | 629419 |
| tr | A0A2I0ME42 COLLI Vitellogenin 2-like OS *Columba livia* OX 8932 GN A306 00005830 PE 4 SV 1 | 189533 | 8.8 | 151.3 | 10 | 127 | 7.7 | 73 | 10 | 54877 |
| **Cockatiel parrot;** *Nymphicus hollandicus* | | | | | | | | | | |
| tr | A0A0Q3PS33 AMAAE Ovalbumin-related protein Y-like protein OS *Amazona aestiva* OX 12930 GN | 45758 | 8.1 | 3217.7 | 16 | 30 | 14.8 | 192 | 8 | 115683 |
| tr | A0A091GPI0 BUCRH Zona pellucida sperm-binding protein 3 Fragment OS *Buceros rhinoceros* si | 32669 | 5.5 | 2184.3 | 2 | 20 | 9.0 | 29 | 2 | 516948 |
| tr | U3JXV8 FICAL Uncharacterized protein OS *Ficedula albicollis* OX 59894 GN LOC101815176 PE 4 SV | 160510 | 9.5 | 1178.0 | 5 | 108 | 4.6 | 91 | 5 | 82838 |
| tr | A0A091IIQ9 CALAN Zona pellucida sperm-binding protein 1 Fragment OS *Calypte anna* OX 9244 | 95792 | 7.1 | 579.4 | 2 | 34 | 3.6 | 16 | 2 | 593785 |
| tr | A0A2I0TKM1 LIMLA Type II cytoskeletal 5-like OS *Limosa lapponica baueri* OX 1758121 GN lla | 63071 | 5.0 | 237.4 | 2 | 55 | 2.5 | 18 | 2 | 528417 |
| tr | A0A2I0THL3 LIMLA Uncharacterized protein OS *Limosa lapponica baueri* OX 1758121 GN llap 16 | 79366 | 5.6 | 129.6 | 2 | 64 | 3.5 | 14 | 2 | 524668 |
| P87498 | VIT1 CHICK Vitellogenin 1 OS *Gallus gallus* OX 9031 GN VTG1 PE 1 SV 1 | 210497 | 9.2 | 43.2 | 2 | 144 | 1.4 | 9 | 2 | 1086 |

PLGS = ProteinLynx Global SERVER

presence of specific proteins (Fig 7). The analysis confirmed the presence of four proteins with a weight of >250 kDa and three proteins weighing approximately 35 kDa in the protein band of VM isolated from the cockatiel parrot eggs. The proteomic analysis confirmed that the 250-kDa protein found in the VM of all the birds analyzed was alpha-2-macroglobulin-like 1 protein, which is an endopeptidase inhibitor or mucin 5B. In the case of partridge and pheasant eggs, this protein was found to be in the lowest quantity. According to the results of proteomic data obtained from ProteinLynx Global SERVER (PLGS), the protein weighing 35 kDa was found as ZP3, which is present in the transparent casing. It is a thick, glycoprotein coat surrounding the oocyte. Such proteins are characterized by variability in their structure. The most conservative glycoprotein is ZP3, which consists of about 400 amino acids. The protein fraction obtained in the case of pheasant and hen eggs weighing 20 and 12 kDa, respectively, showed the presence of type II cytoskeletal keratin proteins present in the protein cytoskeleton. It should be noted that the proteins of the complement system (R7VRC4) were found in the VM of pigeon eggs (Fig 5, S2 Data) but not in the VM of other birds' eggs.

In the case of gray partridge and cockatiel parrot eggs, only three and seven proteins, respectively, were observed (Fig 7). The 15-kDa protein band obtained after the separation of

**Table 3. The proteomic analysis of water-washed vitelline membrane (VM) of selected bird species.** Proteins were identified from the selected bands (according to Fig 5).

| Swiss-Prot/ Trembl accession | Protein | mW (Da) | pI (pH) | PLGS score | Peptides | Theoretical peptides | Coverage (%) | Products | Digest peptides | Protein ID |
|---|---|---|---|---|---|---|---|---|---|---|
| **PRECOCIAL** | | | | | | | | | | |
| **Ring-necked pheasant;** *Phasianus colchicus* | | | | | | | | | | |
| Not determined | | | | | | | | | | |
| **Gray partridge;** *Pedrix pedrix* | | | | | | | | | | |
| Tr | G1MVV5 MELGA Ovotransferrin OS *Meleagris gallopavo* OX 9103 GN TF PE 3 SV 2 | 77727 | 6.6 | 1902.1 | 14 | 72 | 19.3 | 135 | 14 | 498065 |
| O42273 | TENP CHICK Protein TENP OS *Gallus gallus* OX 9031 GN TENP PE 2 SV 1 | 47404 | 5.5 | 734.1 | 1 | 25 | 4.8 | 18 | 1 | 9683 |
| Tr | G1MW54 MELGA Zona pellucida glycoprotein 1 OS *Meleagris gallopavo* OX 9103 GN ZP1 PE 4 SV 1 | 95510 | 7.6 | 623.5 | 4 | 29 | 6.0 | 43 | 4 | 499970 |
| **SUPERALTRICIAL** | | | | | | | | | | |
| **Domestic pigeon;** *Columba livia* | | | | | | | | | | |
| Tr | H0Z0C5 TAEGU Uncharacterized protein OS *Taeniopygia guttata* OX 59729 PE 3 SV 1 | 55310 | 4.8 | 471.5 | 2 | 37 | 3.8 | 21 | 2 | 456330 |
| **Cockatiel parrot;** *Nymphicus hollandicus* | | | | | | | | | | |
| Tr | A0A087R4H2 APTFO Alpha-2-macroglobulin-like 1 Fragment OS *Aptenodytes forsteri* OX 9233 | 106930 | 8.0 | 208.5 | 2 | 61 | 3.2 | 19 | 2 | 336274 |
| Tr | U3JXV8 FICAL Uncharacterized protein OS *Ficedula albicollis* OX 59894 GN LOC101815176 PE 4 SV | 160510 | 9.5 | 207.7 | 4 | 108 | 3.5 | 38 | 4 | 82838 |
| Tr | A0A091FSD8 9AVES Mucin 5B OS *Cuculus canorus* OX 55661 GN N303 00192 PE 4 SV 1 | 233620 | 5.4 | 200.9 | 2 | 140 | 1.4 | 26 | 2 | 652317 |

PLGS = ProteinLynx Global SERVER

the VM of domestic pigeon demonstrated the presence of H0Z0C5 protein, the function of which has not yet been identified.

# Discussion

## VM structure

According to our results, the structure of the VM of superaltricial birds' eggs was much more complex than that of precocial birds' eggs. First, the VM of both pheasant and partridge eggs was composed of three analogous layers: IL, CM, and OL. This finding is interesting because previous studies on the structure of VM of hen eggs have shown the presence of one fibrous IL and one OL separated by a thin continuous layer of CM [1, 25–27]. All three species of birds, namely, hen, pheasant, and partridge belong to the category of precocial birds. This result indicates however the species differences between them. Using the example of VM of hen eggs, Waclawek et al. [28] and Takeuchi et al. [6] previously demonstrated that the IL components are secreted by granulosa cells in the ovarian follicle. According to Bausek et al. [29], at least one of the major IL components—chkZP1—is synthesized in the liver and is transported via the bloodstream to the ovarian follicle. These authors showed that the protein components constituting CM and OL are formed after ovulation in the infundibulum or other parts of the

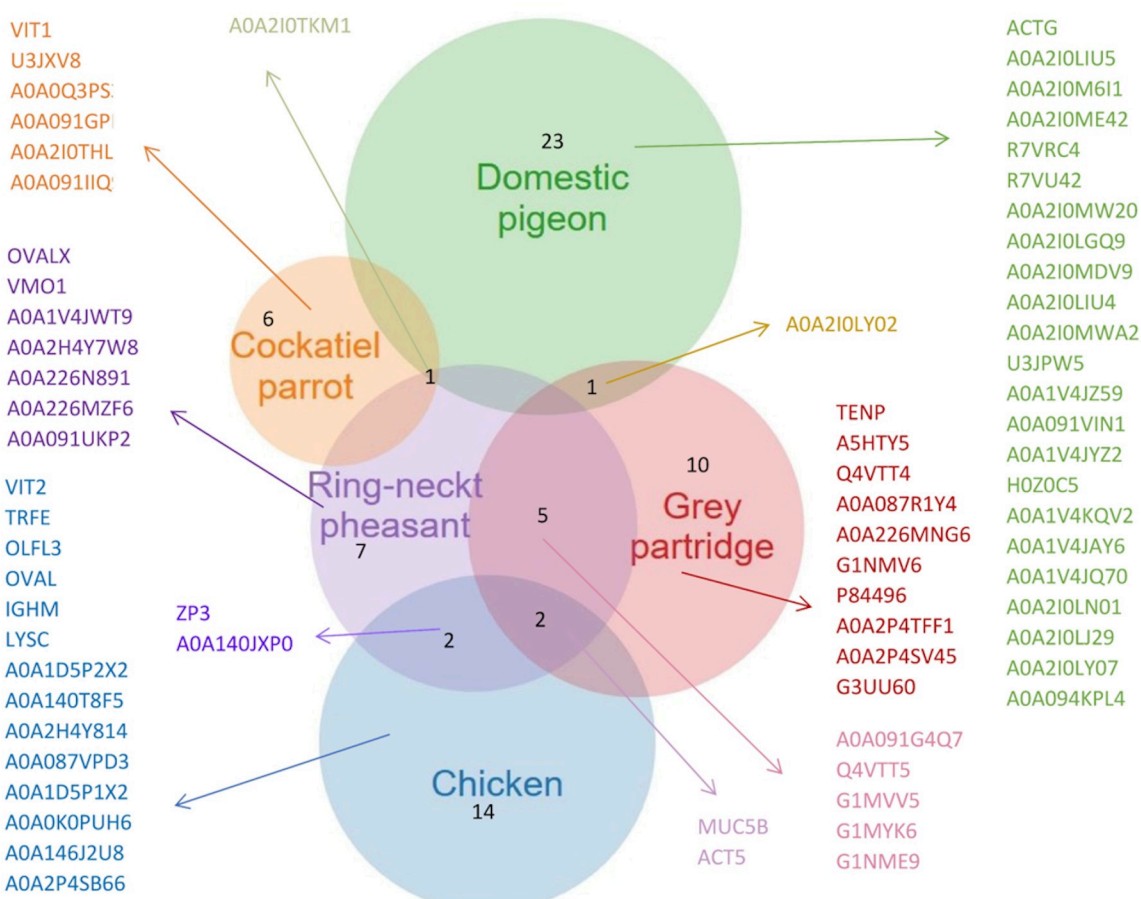

**Fig 6. A representative list of proteins of the vitelline membrane (VM) extracted from the Venn diagrams (a full list is given in Table 2 and S2 Data).**

oviduct during the shift of the yolk sac. However, this cannot be possible in the case of the VM structure of pheasant and partridge because the individual layers are overlapped. Consequently, only $IL_3$ can be formed from the products of the granulosa cells of the ovarian follicle. Both $IL_1$ and $IL_2$ must be produced after ovulation, similar to the three layers of CM and OL. The specification of granulosa cells in the ovarian follicle, including both oocytes and oviduct walls, for glycoproteins producing IL and other proteins such as ovomucin, lysozyme, and VMO-II, which have been characterized as typical for OL, is not excluded [12, 30].

According to the TEM images, the VM of the eggs of superaltricial birds has a much less diversified structure compared to the VM of precocial ones, including that of the hen eggs known from the literature. It is noteworthy that the structure of the VM of pigeon eggs is completely different. By convention, a single layer of IL and OL was marked in the TEM image, but the differences between them are so subtle that the whole structure of VM can be considered as IL formed from multiple thin sublayers. This may be partly due to the fact that the whole VM is made up of flat sheets and not cylindrical fibers as observed in the other species. In the literature, only Chung et al. [7] described the similarity in the structure of the VM of duck and hen eggs. Nevertheless, the structure of the VM of both these species was formed from cylindrical fibers of different thicknesses and less number of sheets and thus appearing similar to the structure observed for pheasant and partridge eggs. Based on these observations,

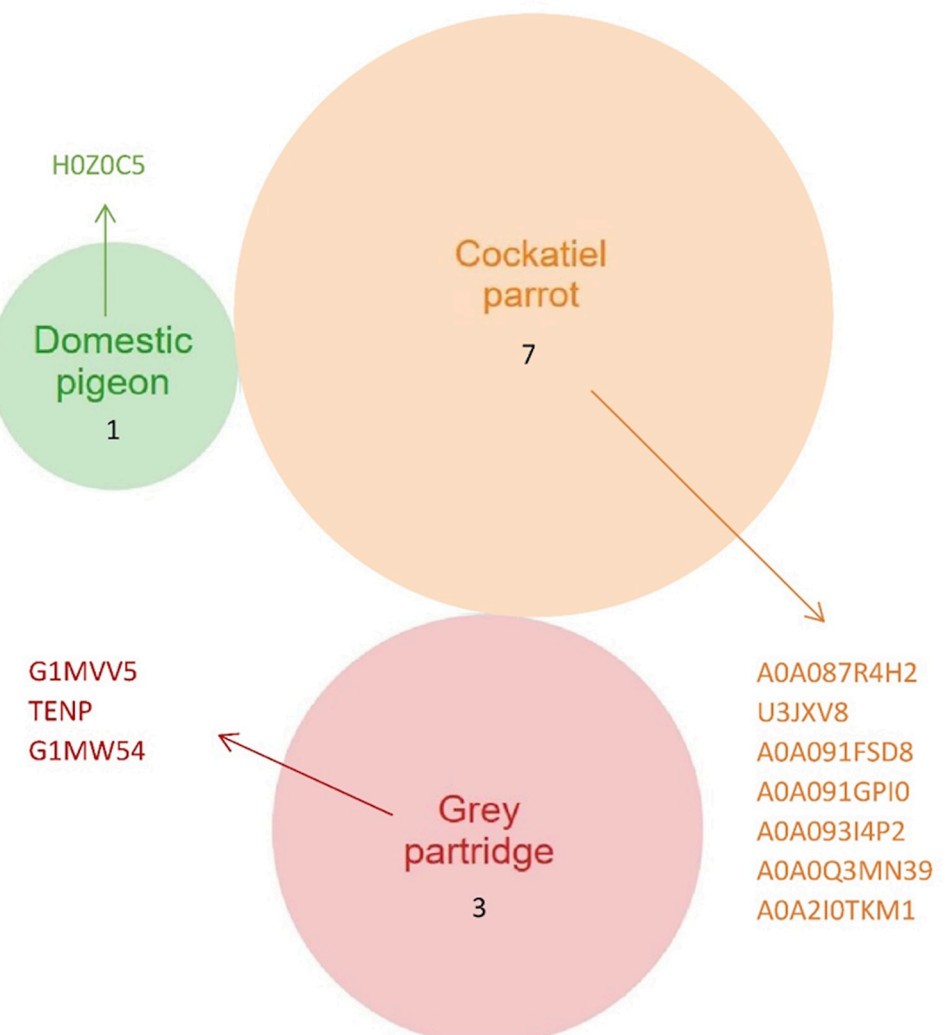

**Fig 7. Species-specific list of proteins of the vitelline membrane (VM) extracted from the Venn diagrams (a full list is given in Table 3 and S3 Data).**

it can be concluded that the different structure of the VM of the pigeon egg is not, however, a characteristic feature of the precocial species, as evidenced by the structure of the VM of parrot eggs, in which case, the VM consisted of three clearly separated layers (as per TEM images)—IL, CM, and OL—and the OL was fibrous and sheet-free structure (as per SEM images), and hence was similar to the VM structure of hen eggs described in the literature. However, compared to the structure of the VM of the hen egg described earlier by Kido and Doi [25], it was completely thin with a much less developed OL.

According to Kido and Doi, structural differences in the VM of the birds of the studied superaltricial and precocial species, and other altricial species for which such a characterization had been performed previously, maybe due to the differences in clutching specificity. First, the functions of the VM can be divided into those that take place during the inhibition of embryonic development before incubation and those that occur during incubation. The embryonic development is inhibited when the egg is laid as a result of temperature reduction from 41°C (body temperature of bird) to ambient temperature. This period differs between the bird

species and depends on the number of eggs in a single clutch and whether the species clutching is synchronized or asynchronized. In the case of species with multiple clutches, in which a strong synchronization is the key hatching strategy, most of the eggs go through a period during which the embryo lives but does not develop while waiting for the incubation to begin. However, the mechanisms of egg "aging" in which the VM plays a significant role constantly occur during this period. The basic mechanism of aging involves the penetration of water through the VM from the egg white to the yolk due to the loss of $CO_2$ in the shell pores and the increase in the pH of the egg white (from 7.6 to 9.7). As a result, the ovomucin–lysozyme complex is disintegrated, and the VM is loosened and becomes more permeable to pathogens [20, 31]. Among the studied species, this threat is incomparably greater for pheasants and partridge eggs. A single clutch consists of 8–12 eggs in the case of pheasant [32] and even up to 20 eggs in the case of partridge [33]. Both these species are characterized by a strong synchronization of clutching, and hence, incubation starts only when the last egg is laid. As a result, the duration between the laying of the first egg and hatching spans even several weeks. On the contrary, pigeons lay only two eggs in a clutch and start brooding as soon as the first one is laid [34]. Parrots use a similar strategy; they lay 4–6 eggs but incubation them after laying the first or second egg and the whole clutch is characterized by asynchronization [35]. Thus, a highly developed VM seen in the eggs of pheasant and partridge may help them in counteracting the negative effects of the long waiting time for the eggs to start incubation. However, it is interesting to explore as to why the structure of the VM of pheasant and partridge eggs is so different from that of the hen eggs presented by other authors, even though they are all precocial birds [1, 25–27]. Analyzing the natural clutching strategies of the wild ancestor of the domestic hen, Red Jungle Fowl, it was observed that its clutch consisted of a small number of eggs (4–6) and the period from the laying of the first egg to the start of brooding did not last longer than 8 days [36]. There is no information available in the literature on the effect of domestication and hen selection on the structure of the VM. Therefore, the structure of the VM of the hen eggs presented by Kido and Doi [25], Tan et al. [26], and Li et al. [27] is probably the same as in their ancestor. Kirunda and McKee [20] also demonstrated that the structure of the VM of a hen egg loosens as early as 7 days after laying, becoming more susceptible to interruption. Seven days is also considered an optimal storage period for the hatching of hen eggs, followed by a significant decline in their biological value [37]. Therefore, it can be assumed that due to a more abundant clutch and a consequently longer period of inhibition of embryo development compared to hens and superaltricial birds, the eggs of pheasant and partridge have a strongly expanded VM. It is assumed to slow down the negative effects of the natural "aging" of eggs.

Apart from a long period of residence in the state of inhibition of embryogenesis, pheasant and partridge eggs are also characterized by a relatively long incubation time, which lasts for about 23–24 days [32, 33]. Despite their high egg weight, the incubation time of hen eggs is short (21 days) [37], whereas the incubation time of the eggs of superaltricial birds is even shorter—18 days in the case of a parrot [35] and only 14 days in the case of a pigeon [34]. During incubation, VM performs both antioxidant and antibacterial functions due to the presence of specific proteins, as well as the mechanical functions favored by its structure. It is also primarily responsible for the transport of nutrients between the yolk and the walls of the blood vessels of the embryo, takes part in the formation of fetal membranes, and enables the sac to be drawn into the body cavity before the clutch [38]. The supply of the substance of the yolk sac in the precocial species is much higher than in the altricial ones, since the chicks of the precocial birds are not fed by the parents, and it must be sufficient for them until they are able to feed themselves. However, the chicks of the superaltricial birds stay in the nest for several weeks and are fed by the parents from the first day, and therefore, they do not need a large supply of yolk material. Thus, maintaining the continuity of the walls of a heavy yolk sac and

pulling it into the body cavity requires a much stronger structure of the VM in the case of precocial birds than that of the superaltricial ones.

## VM proteome

The use of the NanoAcquity Ultra Performance LC (Waters) system combined with a mass spectrometer for proteomic analysis enabled us to conduct a minimalist approach for the processing of the concentrated extract of VM. As a result, it was possible to identify a large number of protein components in the VM of as many as five bird species (Tables 2 and 3 and S2 Data and S3 Data). This is the first comprehensive report on the comparison of the VM proteome of superaltricial and precocial species, which provides a starting point for the determination of the function and the molecular basis of the properties of VM. Most of the components of protein of the VM were previously determined from the other parts of the egg structure (e.g. ovalbumin and ovomucin) [2, 39]. However, the biochemical function of ovalbumin is still undefined. It is most abundant in egg white and is a nonfunctional member of the SERPIN family and has no antimicrobial activity [40, 41]. It probably functions only as a reserve material for a developing embryo. However, ovomucin is responsible for the viscosity of egg white and maintaining its proper structure [42, 43].

In this study, however, proteins with antibacterial activity were mainly identified in the VM of the eggs: ovotransferrin and lysozyme C. Ovotransferrin is an acidic glycoprotein (pI 6.0), which constitutes about 12–13% of the egg white in birds [44]. It inhibits the development of various microorganisms possibly through binding with iron ions, which is an essential growth factor and is responsible for the proper maintenance of the cellular redox status [45]. Lysozyme C is a bacteriolytic enzyme occurring in the form of a monomer and belongs to the group of glycosidic hydrolases. This enzyme may cause damage to the bacterial cell wall membrane system by hydrolyzing the polypeptide bonds [46]. Among the lesser-known proteins, clusterin, alpha-macroglobulin, and olfactomedin were found in the VM of the eggs of the studied birds. Clusterin is a strong ubiquitous extracellular protein that inhibits protein aggregation and precipitation caused by physical or oxidative stress [47]. Studies conducted by other authors showed the presence of clusterin in the shell matrix and white of hen eggs, but it was not described previously as a component of the VM proteome. The function of clusterin and alpha-macroglobulin has been suggested to support the process of egg formation [41]. Olfactomedin plays an important role in dorsal-central modeling during the early embryonic development of hen [48].

OCX-32 was found only in the VM of hen eggs. This protein is secreted in high concentration by the shell gland during the final stage of calcination and is localized mainly on the surface of the shell, forming a cuticle together with mucin [49]. Moreover, this protein belongs to the group of immune proteins with bactericidal characteristics, which increases permeability (bactericidal permeability-increasing protein—BPI) by binding to bacterial lipopolysaccharides. OCX-32 protein was not identified in the VM of the other four bird species, which may indicate that it is a protein that is specific for *G. gallus* species.

Compared to other proteomic studies of different morphological parts of the egg of birds [41], this study identified several new proteins, including vitellogenin 1 (VTG1) and vitellogenin 2 (VTG2). Vitellogenins take part in lipid movement [48]. In addition, these proteins are involved in the biosynthesis of lipovitellins and phosvitin. Phosvitin is an important element of the granular fraction of egg yolk and exhibits antioxidant properties. It also has the ability to chelate metal ions (favorable for the formation of free radicals). Such properties allow an effective inhibition of the oxidation of, for example, phospholipids. Moreover, according to Cordeiro and Hincke [49], these proteins are a source of nutrients for the developing bird embryo.

Another protein that was identified (hen, ring-necket pheasant, gray partridge, cockatiel parrot) in this study is the ZP. It should be noted that this protein was not found in the VM of domestic pigeon eggs. According to the literature, this protein can be located in the whole volume of an egg [50]. The ZP was originally identified in pigs and was named so because it binds to the transparent oocyte casing. It is made up of glycoproteins, and so far, three main glycoproteins of the transparent casing have been identified—ZP1, ZP2, and ZP3. The glycoproteins in the transparent casing are synthesized by oocytes during the growth phase. *N*-Glycans present on the surface of the proteins, especially the high-mannose structures and branched chains of the complex type, play a special role in the binding of the sperm to the transparent casing. These glycoproteins may show a high affinity for the glycoproteins present on the sperm membrane, which consequently suggests that they may be a potential receptor for sperm [51–53]. Ultrastructural studies have shown that the lack of ZP1 prevents the acrosomes from reaching the proper concentration, which results in their fragmentation and interruption of sperm penetration [54]. Moreover, the casing protects the developing embryo until it resides. The second protein (ZP2) was first identified in *Limosa lapponica baueri*, a medium-sized migratory bird belonging to the Sandpipers family. It is noteworthy that this protein was not observed in the VM of the eggs of cockatiel parrot and gray partridge.

Specific VM components that differed between particular bird species included about 10 protein fractions (Fig 4). Most of the proteins were isolated, but their sequences (Tables 2 and 3 and S2 Data and S3 Data) have not yet been adequately characterized in terms of their function. The majority of proteins are similar in the VM of different bird species (they belong to the same group of proteins), but their relative proportions are completely different. The VM proteome of the eggs of hen, ring-necket pheasant, and gray partridge contained proteins that were previously identified (Tables 2 and 3 and S2 Data and S3 Data). These included lysozyme C, ovalbumin, macroglobulins, ZPs, and cytoskeleton-building proteins.

The abundant nature of individual proteins obtained from avian VM demonstrated the association between the structure and the physical traits. In the case of domestic pigeon eggs, which had the highest number of proteins, we found that their abundance could be decisive about the different physical parameters. The proteins that were commonly found in ring-necked pheasant, gray partridge, and cockatiel parrot eggs did not have such a pronounced impact on the structure of the VM. Even in the case of cockatiel parrot eggs, the VM differed considerably in terms of morphology from the VMs isolated from the eggs of other birds. This can be confirmed by the presence of six proteins, which were absent in other birds. The similar structure of VM of ring-necked pheasant and gray partridge, which indicated the presence of five common proteins should be emphasized. However, it should be noted that in-depth proteomic analysis and physical linkages between individual avian membranes should be performed to obtain the highest possible amount of information on the differences between individual bird species. Such knowledge will enable us to identify numerous differences. Furthermore, it will allow noticing individual traits present between bird species, which can be of high significance from the proteome standpoint. Obtaining such information may be the basis for the definition of their functions and properties.

Among the proteins previously reported in the VM proteome, 41 similar fractions were determined. Unfortunately, no single hypothesis can explain the diversity of protein fractions in the VM of the eggs of the studied birds. This suggests that the ecology and life history of birds have most likely changed in the course of evolution. We can only speculate whether there VM constitutes any specific protein, but the complete proteomic data of VM is not yet available. It should be mentioned that the VM database (National Center for Biotechnology Information, ExPASy: SIB Bioinformatics Resource Portal) contains sequences of proteins that are very similar to those playing an important role in the structure of other morphological parts of bird eggs.

## Conclusion

In this study, we demonstrated that the structure of VM differs between different species of birds. There were differences found in the number of VM layers, their course and connections, and in the fibers forming the membrane. Despite the fact that our data is based on a limited number of species, it cannot be definitely confirmed whether these differences are directly related to the nesting specificity of birds (precocial and superaltricial), several observations support the hypothesis. In particular, the structure of the VM of ring-necked pheasant and gray partridge eggs, which are precocial birds, differed from the structure of the VM of cockatiel parrot and domestic pigeon eggs, which are classified as superaltricial birds. The considerable difference in the structure of the VM between cockatiel parrot and domestic pigeon eggs suggests phylogenetic influences. Therefore, future studies on the structure of yolk VM should be conducted considering both the nesting specificity and the phylogenetic classification of bird species.

The proteomic analysis of the VM of precocial birds (cockatiel parrot and domestic pigeon) in relation to superaltricial birds (hen, ring-necked pheasant and gray partridge) showed differences in the presence of proteins characterized by low (<20 kDa) and high molecular weights (>210 kDa). Unidentified proteins were found in all VMs, the function of which has not been completely elucidated. Scientific knowledge on this subject is still inadequate and needs to be broadened by further research and experiments aiming at the meticulous identification of new protein fractions. The analyses conducted in this study showed the presence of protein fractions having an intensity of about 44 and 220 kDa, only in the VM of the precocial species.

This study is the first to report the differences in the protein composition and structure of the VM of precocial and superaltricial birds. The data presented here may broaden the existing knowledge by enabling a better understanding of the protein composition of the VM of birds. In the future, this knowledge of the differences in the structure and protein composition of VM may serve as a tool to identify species on a par with genetic analysis and support systematic differentiation.

## Supporting information

**S1 Data. Raw data for Tables 1 and 2.**
(XLS)

**S2 Data. All the proteins identified in whole VM by SDS-PAGE.**
(XLSX)

**S3 Data. Proteins identified from selected bands (according to Fig 5).**
(XLSX)

## Author Contributions

**Conceptualization:** Krzysztof Damaziak.

**Data curation:** Krzysztof Damaziak, Marek Kieliszek.

**Formal analysis:** Krzysztof Damaziak.

**Methodology:** Krzysztof Damaziak, Marek Kieliszek.

**Project administration:** Krzysztof Damaziak, Mateusz Bucław.

**Validation:** Mateusz Bucław.

**Writing – original draft:** Krzysztof Damaziak.

**Writing – review & editing:** Marek Kieliszek.

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
