## [Decision Letter · Decision Letter 0]

27 Sep 2019

PONE-D-19-16659

Structure and protein identification of some precocial and superaltricial birds eggs yolk vitelline membrane

PLOS ONE

Dear Dr Damaziak,

Thank you for submitting your manuscript to PLOS ONE. After careful consideration, we feel that it has merit but does not fully meet PLOS ONE’s publication criteria as it currently stands. Therefore, we invite you to submit a revised version of the manuscript that addresses the points raised during the review process.

The manuscript has been assessed by 2 reviewers who have requested a number of revisions to improve the manuscript. In particular, Reviewer #2 felt there were significant concerns with the study and requests more data integration and discussion to explain the correlations between structure, physical characteristic and protein composition/abundance. Please revise the manuscript to address all the reviewer's comments in a point-by-point response in order to ensure it is meeting the journal's publication criteria.

We would appreciate receiving your revised manuscript by Nov 09 2019 11:59PM. To enhance the reproducibility of your results, we recommend that if applicable you deposit your laboratory protocols in protocols.io, where a protocol can be assigned its own identifier (DOI) such that it can be cited independently in the future. For instructions see: http://journals.plos.org/plosone/s/submission-guidelines#loc-laboratory-protocols

We look forward to receiving your revised manuscript.

Kind regards,

Miquel Vall-llosera Camps

Staff Editor

PLOS ONE

**Comments to the Author**

1. Is the manuscript technically sound, and do the data support the conclusions?

Reviewer #1: Partly

Reviewer #2: No

2. Has the statistical analysis been performed appropriately and rigorously? 

Reviewer #1: Yes

Reviewer #2: Yes

3. Have the authors made all data underlying the findings in their manuscript fully available?

Reviewer #1: Yes

Reviewer #2: Yes

4. Is the manuscript presented in an intelligible fashion and written in standard English?

Reviewer #1: No

Reviewer #2: No

5. Review Comments to the Author

Reviewer #1: In this manuscript, the authors analyzed the vitelline membranes of precocial and superaltricial birds in aspects of the ultrastructures or protein compositions, and compared the results to discuss whether the characteristics of VM are related to the hatching specification of birds. Zona pellucida, a mammalian homolog of bird VM, is essential for in-vivo fertilization and early embryo development, although the molecular mechanisms that underlie physiological functions of zona pellucida are not clearly resolved. Therefore, further investigations following this study could provide new insights into the zona pellucida-related infertility or disorders of embryo development not only in taxonomic studies of birds. From these point of view, I think this manuscript is worth being accepted after some revision.

I recommend that the authors revise and/or modify manuscript in some points mentioned below.

Major points:

1) lines 19-38

Syntactic complexity or errors throughout in "ABSTRACT" section should be improved.

2) lines 50-52 and 56-57

There might be some inconsistencies in "Introduction" section. I guess the IL is mainly composed of zona pellucida glycoproteins but not collagen.

3) lines 320 and 322

CM should be mentioned and explained in the "Introduction" section and be indicated in the Figure 3.

4) lines 331-335

In chicken and Japanese quail both being included in the precocial birds, either IL or OL appears to be the single layer. According to the authors hypothesis, they should have multiple layers. How to explain the previous observations in chicken and quail VM? Is it possible to explain that the VM of precocial birds are folded after OL formation?

5) lines 493-506

It might be better that "Conclusion" section contains the conclusion bringing together the results from all the experiments in this study not only from the proteomic ones.

Minor points:

1) line 59

"apovitellin" should be replaced with "lipovitellin".

2) lines 259-277

"protein fraction" might be replaced with "protein bands" or "protein signals".

3) line 262

"first line" should be replaced with "first lane".

4) lines 262-263

"egg whites VM" might be replaced with "egg yolks VM".

5) lines 328, 329, 334 and 335

"oocyte infundibulum" or "infundibulum" should be replaced with "ovarian follicle" or "follicle". "infundibulum" is a part of oviduct.

6) line 328

Is "they" mean ZP1? If so, "they" should be replaced for eample with "one of them".

7) line 484

"zona pellucida binding proteins" might be replaced with "zona pellucida sperm binding proteins".

8) lines 494 to 495

Bird species names of the precocial and altrical birds might be exchanged.

Reviewer #2: My comments and suggestions are summarized in the attached file entitled PONE-D-19-16659- 07242019

6. PLOS authors have the option to publish the peer review history of their article (what does this mean?). If published, this will include your full peer review and any attached files.

Reviewer #1: Yes: Hiroki Okumura

Reviewer #2: No

---

## [Author Response · Author response to Decision Letter 0]

22 Oct 2019

Response to the Review 

Manuscript: PONE-D-19-16659

Dear Editor 

We would like to express our gratitude for the assessment of our manuscript entitled „Structure and protein identification of some precocial and superaltricial birds eggs yolk vitelline membrane”. Conducting of the research and preparing of the publication were made by us with the greatest possible precision and care. We would like to point out that for the authors: Krzysztof Damaziak and Marek Kieliszek the affiliations have been changed. This is due to the name changes of our University Faculties from October 1, 2019. Below are corrections for the specific remarks. All the amendments were made to the text and marked with a yellow background. Due to changes in the content of the text, line numbers may not match those before the review:

Reviewer: 1

R: lines 19-38: Syntactic complexity or errors throughout in "ABSTRACT" section should be improved.

A: English in the entire manuscript has been re-verified and corrected by a native speaker. We have attached a suitable certificate in the supplemental materials.

R: Lines 50-52 and 56-57: There might be some inconsistencies in "Introduction" section. I guess the IL is mainly composed of zona pellucida glycoproteins but not collagen.

A: It has been corrected 

R: Lines 320 and 322: CM should be mentioned and explained in the "Introduction" section and be indicated in the Figure 3.

A: In the chapter Introduction we have added the following sentence ‘Between them lies a granular “continuous membrane” (CM – lamina continua) of unreported composition [1].’ We have marked CM in Figure 3 and introduced an explanation in the figure caption.

R: Lines 331-335: In chicken and Japanese quail both being included in the precocial birds, either IL or OL appears to be the single layer. According to the authors hypothesis, they should have multiple layers. How to explain the previous observations in chicken and quail VM? Is it possible to explain that the VM of precocial birds are folded after OL formation?

A: Unfortunately, at the moment we are unable to precisely explain the observations we obtained. The image of three-layer VM of pheasant and partridge was a great surprise to us. We repeated the analyses several times and the TEM image was always similar. It is surprising because in the case of chicken and partridge VM, the structure is indeed single, as pointed out by the reviewer. What is more, we are currently conducting VM analyses of other avian species, including Turdus merula, Turdus philomelos, Rhea americana, Cairina moschata and despite the considerable morphological diversity, VM only cosnsists of single layers. We suspect that this can be related to the evolutionary adaptation of nesting strategies (e.g. high number of eggs per clutch). As we have attempted to explain this in the discussion, the nesting behavior of pheasant and partridge differ from those observed in, among others, wild ancestors of the domestic hen. We suspect, that the answer may be brought by histological tests of the oviduct and VM with the use of various staining types. We will seek to clarify these observations in future studies.

R: Lines 493-506: It might be better that "Conclusion" section contains the conclusion bringing together the results from all the experiments in this study not only from the proteomic ones.

A: We have completed the „Conclusion” as suggested by the reviewer

R: Line 59: "apovitellin" should be replaced with "lipovitellin".

A: We replaced "apovitellin" to "lipovitellin"

R: Lines 259-277: "protein fraction" might be replaced with "protein bands" or "protein signals".

A: It has been corrected

R: Line 262: "first line" should be replaced with "first lane".

A: We replaced "line" to "lane”

R: Lines 262-263: "egg whites VM" might be replaced with "egg yolks VM".

A: We replaced "whites" to "yolks”

R: Lines 328, 329, 334 and 335: "oocyte infundibulum" or "infundibulum" should be replaced with "ovarian follicle" or "follicle". "infundibulum" is a part of oviduct.

A: It has been corrected

R: Line 328: Is "they" mean ZP1? If so, "they" should be replaced for eample with "one of them".

A: In the indicated fragment we have introduced the following sentence: “According to Bausek et al. [29] at least one of the major IL components - chkZP1 is synthesized in the liver and is transported via the bloodstream to the ovarian follicle.”

R: Line 484: "zona pellucida binding proteins" might be replaced with "zona pellucida sperm binding proteins".

A: It has been replaced

R: Lines 494 to 495: Bird species names of the precocial and altrical birds might be exchanged.

A: It has been corrected

Reviewer: 2

R: The authors aimed to correlate the protein and structure specificities of the vitelline membrane to these brooding/hatching characteristics. Although the concept is very interesting, the article deeply lacks convincing/striking conclusions and correlations between experimental observations are either missing or too hazardous.

A: We agree with the reviewer that the conclusions and discussion may be dangerous. However, this is the first report describing possible differences resulting from the structure and protein content in the VM of individual bird species. We've improved results discussions to provide new information. At the same time, we described the Venn diagram that will make the description of the results more attractive.

R: Moreover, the article needs to be edited by an native English speaker (grammatical errors + sentences to be rephrased) and to my opinion, it doesn’t fulfill quality requirements for a research article: figure legends are in the main text (at the end of each paragraph)

A: The article was corrected by Native English Speaker (www.translmed.com). Figures and legends have been included in the text in accordance with the requirements of the magazine PlosONE.

R: The SEM and MEB pictures are of poor quality and related figures need to be clarified and detailed in related legends. Some data related to chicken egg are missing in Table 1 and in SEM/TEM data considering that you performed proteomic analysis of VM from chicken eggs. The number of proteins identified by mass spectrometry is very low and it is very difficult to perform conclusions as the tables are quite confusing. A Venn diagram comparing results obtained from each species (number of proteins per species, common and specific proteins) would have been clearer.

A: TEM and SEM images were prepared in line with the PLOS ONE journal guidelines. We perceive them as a very high quality - clear and sharp. Their quality is deteriorated after sending them to the journal and PDF generation. Unfortunately, this is beyond our control. In line with the reviewer’s suggestion, we have developed two Venn diagrams: Figure 6 in which we have presented the amount of proteins defined for individual species and the common proteins. Figure 7 presenting specific proteins defined based on the selected bands. This study did not analyze the VM structure of chicken egg yolk with the use of SEM and TEM. This stems from the fact that such studies have been conducted and published on numerous occasions. Therefore, we assumed that this kind of information is well known and one can use the available knowledge with full confidence. Restriction of the analysis by excluding the hen VM structure analysis, which is expensive, labor-intensive and time-consuming, enabled us to better focus on VM analyses of the remaining 4 species. This was highly important for us, as none of the described analyses has ever been performed for either ring-necked pheasant, grey partridge, cockatiel parrot, or domestic pigeon. However, we decided to perform protein analyses for hen VM as such data have been published only on several occasions, and we were aiming at a comparison. In the study of protein identification, we have treated hen VM as reference and bridge between our results and the literature knowledge.

R: Figures/pictures are of poor quality and are not clear to readers.

A: All Figures/pictures have been prepared in line with the PLoS ONE Author Guidelines. Their deteriorated quality stems from processing and PDF file development in the journal's system. This is beyond author's control. The minimum photo resolution is 300x300 DPI.

R: Abstract: Please clearly indicate the name of the bird species studied and compared. The number of proteins identified for each species + common and specific proteins have to be mentioned. Replace “proteomic structure” by “proteomic composition or pattern”

A: We have added full avian species names in the abstract. We have provided the amount of proteins identified per species. We have not mentioned all proteins common between individual species because this prohibited by the editorial limitation of the word count for this chapter. This has been described in detail in the results chapter based on the attached Venn diagrams. We have replaced “proteomic structure” with “proteomic composition or pattern”. However, we suggest a complete deletion of this sentence. In our opinion it is not necessary in the abstract and we may not exceed the number of 300 words.

R: Introduction: Line 48. The vitelline membrane is the matrix for yolk sac expansion over the yolk.

A: It has been corrected

R: Introduction: Line 51. IL components are expressed by the liver of laying hens but also granulosa cells

A: The text has been completed

R: Introduction: Line 62. OCX36 is not specific to the VM but to the eggshell. Apolipoprotein is a yolk-protein. Proteins that seems to be more specifically found in the vitelline membrane are VMO-I and VMO-II (also known as AvBD11)

A: A: Of course, apolipoprotein is a yolk protein but it has also been identified in the hen's egg membrane (Mann, 2008). It should be noted that depending on the species of bird, the content of this protein (apolipoprotein) may be different in the vitelline membrane.

We agree with the reviewer. Of course (VMO-I and VMO-I) these are the most popular proteins found in vitelline membrane.

Research presented by Gautron (2011) showed that the OCX36 protein is eggshell-specific protein that is secreted by the regions of the oviduct responsible for eggshell formation. It has also been identified in vitelline mebrane.

Mann, K. (2008). Proteomic analysis of the chicken egg vitelline membrane. Proteomics, 8(11), 2322-2332.

Gautron, J., Rehault-Godbert, S., Pascal, G., Nys, Y. & Hincke, M. T. (2011). Ovocalyxin-36 and other LBP/BPI/PLUNC-like proteins as molecular actors of the mechanisms of the avian egg natural defenses. Biochemical Society Transactions, 39(4), 971-976, doi:10.1042/BST0390971

R: Introduction: It would be more convincing to explain the difference in VM structures/composition by phylogenetic analyses rather than on their affiliation to precocial and superaltar birds. Such a hypothesis would be better in the discussion and further prospects.

A: Our aim was to demonstrate differences depending on the nesting specificity of birds. Analysis of structure and composition of VM through phylogenetic analysis would be indeed highly interesting and we will seek to perform it in the future. However, it seems that it is too early for such an analysis. The available literature lacks information on VM of other birds than chicken, partridge and duck. Our team has recently expanded the available knowledge on the analysis of VM ratite birds (ostrich, emu and rhea). Now we have selected 4 further species for which the VM analyses are performed for the first time. We are currently analyzing the structure and protein composition of VM of Turdidae species. We hope that soon we will be able to conduct a phylogenetic meta-analysis. In the Material and methods chapter we have supplemented the information on the classification of birds to given families, but we would like to keep the main objective of the study without major changes.

R: Introduction: Please indicate that Gallus gallus, Perdix Perdix and Pahsioanus colchicus are galliforms, Nymphicus hollandicus is a psytaciform and Columba livia a columbiform.

A: We have supplemented the information. We have introduced names of systematic orders but we believe that this will be more appropriate in the M&M chapter. Of course, we agree with the reviewer this is the next stage (with more different species of birds) of the research we carry out. In this work we have added a special chart: Venn diagram - a diagram to illustrate the relationship between sets. 

R: Material and methods: Egg collection. Indicate the strain of the laying hens used form proteomics.

A: Data has been corrected. Information about the hen strain (ISA BROWN) from which eggs were used analyzes was supplemented.

R: Material and methods: Scanning electron microscopy. Please insert a specific paragraph for VM preparation since these preparations were also used for proteomics.

A: We have inserted a separate chapter on VM preparation. 

VM samples for SEM analysis were prepared following the methodology described by Kirund and McKee [20]. After breaking the shell, the egg content was poured onto a separator to separate the egg yolk from the white. After weight determination (± 0.1 g), yolk was placed on a glass pan so that the germ disc was visible on the surface. VM was cut with a scalpel around the egg yolk about halfway up. The VM was then rinsed in deionized water (~4°C) until all residues of the yolk visible to the naked eye were removed The weight of the whole VM (±0.1 mg) was determined and then the area of the embryonic disc (not analyzed area) was separated.

Protein identification is presented in a separate chapter: Protein identification

R: Material and methods: TEM. Indicate what Epon 182 is. Indicate the number of biological replicates

A: Eight biological replicates were performer.

R: Material and methods: Protein extraction and gel electrophoresis. Line 145. I do not understand the reason for using “15 μL of enzymatic proteins”. What is “enzymatic” ?

A: This is a mistake in translation. Of course, there should be only "proteins".

R: Results: Eggs morphology: Table1. use VM thickness instead of width. How did you get this value ? From SEM/TEM data ? Please indicate.

A: The VM thickness was measured on TEM images via Nis Elements version 5.10, Nikon optical microscope (type 104c, Japan).

R: Results: VM Structures. Do not distinguish between precocial and superaltrical birds here. These are results not discussion, please use the respective name of birds (chicken, pheasant, partridge, parrot and domestic pigeon). The use of precocial and superaltrical words may be confusing for non experts.

A: This has been corrected

R: Results: Line 219. Remove this legend at the end of the article + add details about magnification.

A: Figure legends are introduced in the text in place for the proposed presentation. They have been introduced in this manner because such are the requirements of the PLoS ONE Submission Guidelines. 

„Figure captions

Figure captions must be inserted in the text of the manuscript, immediately following the paragraph in which the figure is first cited (read order). Do not include captions as part of the figure files themselves or submit them in a separate document.”

All magnifications SEM and TEM are present in the photographs with automatic caption. We understand that they were not visible due to the quality change of these files. Properly formatted micrograms are supplied to the editorial office, thus we believe that if the manuscript is published, they will be clearly visible.

R: Results: On the figures 1, please indicate A, B, C for each panel and their meaning with respective species to facilitate the reading. . What is the third panel? indicate it in the legend (it appears only in the text).

A: Photos must have been formatted incorrectly when building the PDF. In each photo, information is added: precocial or superaltrical. All photos have a resolution of 300x300 DPI.

R: Results: VM Proteome. Fig. 5. Remove 30 μL from the legend. This is not informative. The quantity 80 μg at the end of the legend is informative

A: It has been removed

R: Results: 287. Please start with the number of proteins per species with a Venn diagram showing common and specific proteins between species. Thus you need to perform blast and alignment analyses to identify homologous proteins that may have different protein names depending on the species.

A: This has been corrected. We have changed the discussion of the results obtained. We also made a Venn diagram.

The conducted characteristics of the proteins in the vitelline membrane with the use of Venn diagram (Fig. 6) for hen demonstrated 14 proteins, which have not been determined in the VM of all birds. In the case of pheasant VM 7 proteins were identified, which were absent from other birds. It should be emphasized that the protein structure of pheasant VM was most closely related in terms of the presence of proteins present in the VM of other birds as well. The most pronounced similarity of pheasant VM was determined for grey partridge (5 proteins) and chicken (2 proteins). The lowest number of proteins (6) among the analyzed avian VM was found in cockatiel parrot. It should be emphasized, that the presence of the same protein (A0A2IOTKM1) was found for cockatiel parrot, ring-necked pheasant and pigeon. Domestic pigeon VM analysis demonstrated the highest number of protein bands (23).

In addition, individual protein bands obtained through electrophoretic separation (Fig. 5) were selected in the study and were subjected to an in-depth analysis in terms of the presence of specific proteins (Fig. 7). The conducted analysis demonstrated the highest number of 4 proteins in the case of the protein band (>250 kDa) of VM isolated from cockatiel parrot egg and 3 proteins from the protein band with the weight of approx. 35 kDa.

R: Results: You infer that alpha2-M like 1 is specific to the parrot but at least in chicken you also have an alpha-2 M protein (the last protein of the chicken section in the table) but also in the pheasant (A0A091G4Q7) etc.

A: Thank you to the reviewer for the information. The data has been corrected. We have deleted invalid data. This protein was also confirmed in an additional analysis. Supplementary S3 Data file.

R: Discussion: The discussion has to be rewritten to better show the correlation between structure, physical characteristic and protein composition/abundance.

A: A: In the discussion, chapters about proteomics and physical features were written in separate chapters. After all, we've added new information. We believe that the combination of all this information could affect the occurrence of chaos and inconsistencies while reading the article. We have determined that such a division will be the most appropriate.

The abundance character of individual proteins obtained from avian VMs demonstrated association with the structure and individual physical traits. In the case of pigeon, for which the highest number of proteins was obtained, we determined that their abundance could be decisive for its different physical parameters. The identified single common proteins that have also been found in parrot, pheasant and partridge did not have such a pronounced impact on the similar structure towards the tested VM membranes. In the case of the structure of parrot VM, it also differed considerably in morphological terms from other VMs isolated from bird eggs. This can be confirmed by the presence of 6 proteins, which were absent from other birds. The similar structure of partridge and pheasant VM should be emphasized, which may indicate the presence of 5 common proteins. However, it should be emphasized that an in-depth analysis of proteomic and physical linkages between individual avian membranes should be continued to obtain the highest possible amount of information on the differences occurring between individual bird species. Such knowledge will enable us to understand and reveal numerous differences. Furthermore, it will allow noticing individual traits present between bird species, which can be of high significance from the proteome standpoint. Obtaining such information may be the basis for the definition of their functions and properties.

R: Discussion: You may have the same overall protein composition with some subtle difference in sequence but what make the VM structure different may results from proteins interaction that may differ between species and relative abundance.

A: Of course, we agree with the reviewer's suggestion. Protein interactions occurring in the VM membrane can change its structural structure. Determination of individual protein sequences and their conformations in the PyMol program and their impact on the VM structure - this is the topic for the next scientific article that we will prepare in the future. We want to perform an in-depth analysis of individual VM membranes. Many proteins have not yet been fully characterized and their function is still unknown. This is a new topic that will allow you to deepen knowledge about the construction of the VM and its role.

R: Discussion: The presence of “yolk” proteins such as vitellogenins, apolipoproteins” and white proteins (ovalbumin, ovomucin) may rather reflect some yolk/white contamination during VM preparations. Depending on the species, the viscosity of the white may be different and stickier. The conclusions are quite hazardous in this paragraph.

A: We agree with the reviewer that the conclusions may be dangerous. However, we think this is the first report on the analysis of proteins isolated from the VM of various birds. Vitellin membrane has been very carefully prepared. Several membrane extractions were performed.

R: Line 337. VMO-II is written three times.

A: It has been corrected

---

## [Decision Letter · Decision Letter 1]

22 Nov 2019

PONE-D-19-16659R1

Structure and protein identification of some precocial and superaltricial birds eggs yolk vitelline membrane

PLOS ONE

Dear Dr. Damaziak,

Thank you for submitting your manuscript to PLOS ONE. After careful consideration, we feel that it has merit but does not fully meet PLOS ONE’s publication criteria as it currently stands. Therefore, we invite you to submit a revised version of the manuscript that addresses the points raised during the review process.

One of the reviewers suggested rejection so this Academic Editor did not send the revised version back to him/her.  Instead, this Academic Editor served as the second review.  Please see my comments below.  

PONE-D-19-16659R1

The authors should be commended that they made major efforts in improving the English of the manuscript from professional editing and comments of the reviewers.  The manuscript, however, is still very rough in both description and English expression.  These deficiencies reduce the readers’ ability to understand the data/findings and undermine the perception of the importance of the work.  They may also likely induce the conception of poor work quality by the authors because readers only see the manuscript not the actual work.  It is, therefore, to the author’s best interest to present the best manuscript possible.  With all these said, the study encompasses large amount of electron microscopy and proteomics work.  The information presented will increase our understanding of the diversity of nature.  The data are therefore worthy of being seen by the scientific community and the general public.

The expertise of this Academic Editor is not in the field of Avian reproduction.  I therefore provided comments to improve the understanding of the manuscript/data from the view of an outsider.

Please change the title to: Structure and protein characterization of the egg yolk vitelline membranes of precocial (common names of the birds???) and superaltricial birds (??)

L23: “the species”: what species?

Also please change the statement to “we analyzed how the structure and protein composition of vitelline membrane (VM) differ among ?? species”.

L27: please change “enable to” to “be important for counteracting”. What do you mean by “complex” some specificity should be given here.

L29: “triple and three-layer”: are these different things or just redundancy?

L30: what is the difference between VM and VM sheets? If not, please use one term consistently throughout the text.

L32: are the results for all birds or for one type? Please specify. Please change “weights” to “molecular weights”. What are protein fractions? How are they fractioned?

L37-39: too much repeats.  Please delete them.

L56: Please change the first “is” to “of”.

L59: please change “so far” to “previously”.

L63: what do you mean by “which are known currently”? Each one of them has been identified?

L64: please remove “first” unless you want to say that they have been found elsewhere later and therefore no longer specific to VM.

L67: Please remove “available from studies”

L70: Please add “the” before “hen”.  When referring an animal as a species, please either use “hens” or “the hen”.  There are similar mistakes in the rest of the text.  Please change them all.

L72: what is the course of the fibers? Do you mean pattern?

L73: “Additional structures” such as ???

L76: “offspring ones”? do you mean “offspring”?

L77: “the authors”: are you still referring those that published the studies you mentioned earlier? From the context, “the authors” in L71 referred to them.

L80: Please change “The birds” to “Birds”. Please also add an explaining phrase after “precocial” (just like you did for the other three categories) to be parallel and coordinated.

L83: please change “first days” to “first few days”.

L85: please change “the chicks” to “chicks”.

L88: using % for egg yolk and water does not make sense.  Because water is part of the yolk.  At least change the “a large proportion” to “a high percentage” so people are not misled into thinking you are talking about the two proportions of the eggs.

L93: “a few “ and “several” are both unspecified and don’t differ much.  I suggest you just use one of them.

L94: please change “storage” to “delayed incubation”. By “storage” you gave the impression that the eggs were taken and stored deliberately.

L98: Please add “using ?? and ?? as models” before “We”.

L112: Please remove “to be” and add “because fertile males were present” (if this is the case).

L181: molecular weights?  Please change all such occurrences in the following sections.

L183: please remove “used”.

At the ends of the sections for SEM, TEM and Protein identification, please add a statement of the number of samples used.  For example, “Ten eggs from each of the 4 species were analyzed yielding a total samples of 40”.

L190: please change to “Egg and VM characteristics”

L191: Please change “morphological traits” to “weight characteristics”.  There were no morphology characterization.

Table 1: please change “Results of the comparative analysis of the morphological” to “Weights”.

Fig 3: From the labeling it appears that CM is a very thin layer of membrane with little characteristics to be seen from the figure.  What did the authors use to identify this layer?  Please describe its features and characteristics.

L246-247: “All the three layers were also characterized by a layered structure”.  This statement does not add anything unless the authors did not state it clearly.  They have already described in previous statements that the VM were triple-layered.

In the text, “Fig” was used but in the figure legend, “Fig.” was used, please be consistent and I suggest Fig. be used because it is abbreviated.

Fig 4 was not provided in the complied file.

L254: please change “corresponded to” to “formed”

L278: Please change “in” to “for”, please also remove “of proteins”

L298-300: Fig 5 legend. Please explain what the red arrows are. 

L305-317: please move this section to after Table 3

L318: what bands were selected?  Are those the red arrowed ones?  Please specify.

L319: please remove “in the study”

L322 and 324: 4 proteins >250 kDa and 3 proteins of 35kDa were found.  Later only one of each were described.  Please explain the others.

L325: please add “ZP3” here

L310-311: In Fig 5, proteins of the complement system were not noted.  Which ones are referred to here?

L333: similarly, 3 proteins were observed but only one was described.  What happened to the other two?

Tables 2 and 3: please change pH, PLGS score, and % of coverage to contain one decimal point.  Please change the molecular weight to kDa to be consistent with the text.  Also in both tables for most protein entries, portions of the IDs were presented twice consecutively. For example: A0A2P4SB66 A0A2P4SB66.  What is the reason for this annotation?

Table 3: please indicate which section corresponds to which red arrow.  If all VM proteins were identified and presented in Table 2 (as indicated in the title of Table 2), why list them again in Table 3?  Are these presented for a second time?  If so, there is no need to have this Table.

Please note that this Academic Editor did not have sufficient time to review the discussion section.  Please use the above editing as a guideline and make revision on the discussion.

We would appreciate receiving your revised manuscript by Jan 17, 2020. To enhance the reproducibility of your results, we recommend that if applicable you deposit your laboratory protocols in protocols.io, where a protocol can be assigned its own identifier (DOI) such that it can be cited independently in the future. For instructions see: http://journals.plos.org/plosone/s/submission-guidelines#loc-laboratory-protocols

We look forward to receiving your revised manuscript.

Kind regards,

Xiuchun Tian

Academic Editor

PLOS ONE

Reviewers' comments:

Reviewer's Responses to Questions

**Comments to the Author**

1. If the authors have adequately addressed your comments raised in a previous round of review and you feel that this manuscript is now acceptable for publication, you may indicate that here to bypass the “Comments to the Author” section, enter your conflict of interest statement in the “Confidential to Editor” section, and submit your "Accept" recommendation.

Reviewer #1: (No Response)

2. Is the manuscript technically sound, and do the data support the conclusions?

Reviewer #1: Yes

3. Has the statistical analysis been performed appropriately and rigorously? 

Reviewer #1: I Don't Know

4. Have the authors made all data underlying the findings in their manuscript fully available?

Reviewer #1: Yes

5. Is the manuscript presented in an intelligible fashion and written in standard English?

Reviewer #1: Yes

6. Review Comments to the Author

Reviewer #1: Compared to the previous version of manuscript, I think the revised version is much more sophisticated as a whole. Although there are many topics that are still remain unclear for example in the formation mechanism(s) of the structurally diverse VL among bird species, the authors will raise problems properly with this manuscript.

I found only one minor point as below.

1. "after ovulation in the ovarian follicle" in Line 373 in the corrected manuscript should be corrected to "after ovulation in the infundibulum".

7. PLOS authors have the option to publish the peer review history of their article (what does this mean?). If published, this will include your full peer review and any attached files.

Reviewer #1: No

---

## [Author Response · Author response to Decision Letter 1]

15 Dec 2019

Response to the Reviewers 

Manuscript: PONE-D-19-16659R1

Dear Editor,

We would like to express our gratitude for assessing our manuscript entitled “Structure and protein identification of some precocial and superaltricial birds eggs yolk vitelline membrane.” We conducted the research and prepared the manuscript with the greatest possible precision and care. We have stated the corrections for the specific remarks below. All the amendments have been made to the text and highlighted in yellow. Due to changes in the content of the text, the line numbers may not match those in the unreviewed manuscript.

R: Please change the title to: Structure and protein characterization of the egg yolk vitelline membranes of precocial (common names of the birds???) and superaltricial birds (??)

A: We have changed the title as per the suggestion.

R: L23: “the species”: what species?

Also please change the statement to “we analyzed how the structure and protein composition of vitelline membrane (VM) differ among ?? species”.

A: We have corrected the statement.

R: L27: please change “enable to” to “be important for counteracting”. What do you mean by “complex” some specificity should be given here.

A: We have modified the statement.

R: L29: “triple and three-layer”: are these different things or just redundancy?

A: We have removed the words “and three-layer.”

R: L30: what is the difference between VM and VM sheets? If not, please use one term consistently throughout the text.

A: We have removed this part of the sentence from the abstract, as it shall not provide any explanation. Sheets are the second structure found following fibers in the OL VM. This has also been discussed in an earlier study of Chung et al. (2010). We have mentioned this in the Results and Discussion (VM structure) sections.

R: L32: are the results for all birds or for one type? Please specify. Please change “weights” to “molecular weights”. What are protein fractions? How are they fractioned?

A: “We found the number of protein fractions to vary from 19 to 23, with molecular weights in the range of 15–250 kDa, depending on the species”—this information pertains to different species of birds.

Protein fractions are individual protein vacuums that are obtained after the SDS-PAGE electrophoresis and have different molecular weights (kDa).

SDS electrophoresis fractionates the proteins according to their mass. This type of electrophoresis is currently the most commonly used and can be applied either individually as an analytical method or as a part of a series of further, more complex studies (e.g. two-dimensional). Fractionation occurs depending on the length of the polypeptide chain, and you can determine the mass of a given protein by comparing with the appropriate standards. This method allows determining the protein mass with an accuracy of 5–10%.

R: L37-39: too much repeats. Please delete them.

A: We have changed the sentence to avoid repetition.

R: L56: Please change the first “is” to “of”.

A: We have corrected the word.

R: L59: please change “so far” to “previously”.

A: We have corrected as per the suggestion.

R: L63: what do you mean by “which are known currently”? Each one of them has been identified?

A: Yes. They are presented in the study of Mann (2008) as “the most comprehensive dataset available at present and complements proteomic analyses of chicken vitelline membrane compartments published previously.”

Mann K. Proteomic analysis of the chicken egg vitelline membrane. Proteomics. 2008; 8: 2322–2332. https://doi.org/10.1002/pmic.200800032 PMID: 18452232.

R: L64: please remove “first” unless you want to say that they have been found elsewhere later and therefore no longer specific to VM.

A: We have removed the word.

R: L67: Please remove “available from studies”

A: We have deleted this text.

R: L70: Please add “the” before “hen”. When referring an animal as a species, please either use “hens” or “the hen”. There are similar mistakes in the rest of the text. Please change them all.

A: All language errors have been corrected by a Native Speaker English (Translmed Publishing Group).

R: L72: what is the course of the fibers? Do you mean pattern?

A: We have substituted the word “course” by “pattern.”

R: L73: “Additional structures” such as ???

A: These structures do not have a name. They appear in the form of tabs, but this is not an official term. In the cited literature, they are depicted in the SEM micrographs.

R: L76: “offspring ones”? do you mean “offspring”?

A: We have deleted the word “ones.”

R: L77: “the authors”: are you still referring those that published the studies you mentioned earlier? From the context, “the authors” in L71 referred to them.

A: We have corrected the sentence.

R: L80: Please change “The birds” to “Birds”. Please also add an explaining phrase after “precocial” (just like you did for the other three categories) to be parallel and coordinated.

A: We have provided an explanation in the sentence.

R: L83: please change “first days” to “first few days”.

A: We have corrected as per the suggestion.

R: L85: please change “the chicks” to “chicks”.

A: We have corrected as per the suggestion.

R: L88: using % for egg yolk and water does not make sense. Because water is part of the yolk. At least change the “a large proportion” to “a high percentage” so people are not misled into thinking you are talking about the two proportions of the eggs.

A: We have removed the information on water content in eggs.

R: L93: “a few “ and “several” are both unspecified and don’t differ much. I suggest you just use one of them.

A: We have corrected the sentence.

R: L94: please change “storage” to “delayed incubation”. By “storage” you gave the impression that the eggs were taken and stored deliberately.

A: We have changed the word.

R: L98: Please add “using ?? and ?? as models” before “We”.

A: We have changed the sentence as follows:

“We also identified the proteins present in VM using the NanoAcquity Ultra Performance LC (Waters) system.”

R: L112: Please remove “to be” and add “because fertile males were present” (if this is the case).

A: We have changed the text.

R: L181: molecular weights? Please change all such occurrences in the following sections.

A: We have corrected as per the suggestion.

R: L183: please remove “used”.

A: We have removed the word.

R: At the ends of the sections for SEM, TEM and Protein identification, please add a statement of the number of samples used. For example, “Ten eggs from each of the 4 species were analyzed yielding a total samples of 40”.

A: We have corrected the data as follows:

“Six eggs from each of the 4 species were analyzed yielding a total sample of 24.”

R: L190: please change to “Egg and VM characteristics”

A: We have changed the subtitle.

R: L191: Please change “morphological traits” to “weight characteristics”. There were no morphology characterization.

A: We have changed the statement.

R: Table 1: please change “Results of the comparative analysis of the morphological” to “Weights”.

A: We have changed as per the suggestion.

R: Fig 3: From the labeling it appears that CM is a very thin layer of membrane with little characteristics to be seen from the figure. What did the authors use to identify this layer? Please describe its features and characteristics.

A: We agree that CM is poorly visible, and therefore, we did not determine it in the SEM images in the first version of our manuscript. However, one reviewer suggested that CM can be determined, as has been done in earlier studies. CM forms a very thin membrane between the IL and OL. Using an example, we would like to show the reviewer how CM has been determined in other studies, which were used as a model for ours. The below scan presents the CM determined in the publication of: Kido S, Doi Y. Separation and properties of the inner and outher layers of the vitelline membrane of hen’s eggs. Poult Sci. 1988; 67: 476–486. https://doi.org/10.3382/ps.0670476.

R: L246-247: “All the three layers were also characterized by a layered structure”. This statement does not add anything unless the authors did not state it clearly. They have already described in previous statements that the VM were triple-layered.

A: We have deleted the sentence.

R: In the text, “Fig” was used but in the figure legend, “Fig.” was used, please be consistent and I suggest Fig. be used because it is abbreviated.

A: We have changed “Fig” to “Fig.” consistently in the manuscript.

R: Fig 4 was not provided in the complied file.

A: We have provided Fig. 4.

R: L254: please change “corresponded to” to “formed”

A: We have changed the word.

R: L278: Please change “in” to “for”, please also remove “of proteins”

A: We have changed as per the suggestion.

R: L298-300: Fig 5 legend. Please explain what the red arrows are.

A: We have explained this in the caption of Fig. 5.

R: L305-317: please move this section to after Table 3

A: We have moved the entire text below Table 3.

R: L318: what bands were selected? Are those the red arrowed ones? Please specify.

A: Yes. We have provided this information in the text.

R: L319: please remove “in the study”

A: We have removed the text.

R: L322 and 324: 4 proteins >250 kDa and 3 proteins of 35kDa were found. Later only one of each were described. Please explain the others.

A: Proteomic analysis confirmed that the 250-kDa protein found in the VM of all the birds analyzed was alpha-2-macroglobulin-like 1 protein, which is an endopeptidase inhibitor or mucin 5B. In the case of partridge and pheasant, this protein was observed at the lowest intensity.

R: L325: please add “ZP3” here

A: We have completed the sentence.

R: L310-311: In Fig 5, proteins of the complement system were not noted. Which ones are referred to here?

A: In Fig. 5, we present the entire protein profile of the VM of individual cacti (supplementary materials). For further detailed proteomic analysis, we selected the individual numbered protein fractions.

R: L333: similarly, 3 proteins were observed but only one was described. What happened to the other two?

A: In the case of gray partridge and cockatiel parrot, only three and seven proteins were observed, respectively (Fig. 7). The 15-kDa protein band obtained after the separation of the VM of domestic pigeon demonstrated the presence of H0Z0C5 protein, the function of which has not yet been identified.

The data are presented in the supplementary materials and the Venn diagram.

R: Tables 2 and 3: please change pH, PLGS score, and % of coverage to contain one decimal point. Please change the molecular weight to kDa to be consistent with the text. Also in both tables for most protein entries, portions of the IDs were presented twice consecutively. For example: A0A2P4SB66 A0A2P4SB66. What is the reason for this annotation?

A: We have corrected everything and removed the IDs that were repeated by mistake.

R: Table 3: please indicate which section corresponds to which red arrow. If all VM proteins were identified and presented in Table 2 (as indicated in the title of Table 2), why list them again in Table 3? Are these presented for a second time? If so, there is no need to have this Table.

A: The red arrows represent the protein fractions that were selected for the detailed proteomic analysis of the VM birds. The detailed analysis of these fractions is presented in the supplementary material S3. We believe that all the results should remain in the article.

R: "after ovulation in the ovarian follicle" in Line 373 in the corrected manuscript should be corrected to "after ovulation in the infundibulum".

A: We have changed the text.

---

## [Decision Letter · Decision Letter 2]

31 Dec 2019

PONE-D-19-16659R2

Characterization of structure and protein of vitelline membranes of precocial (ring-necked pheasant, gray partridge) and superaltricial (cockatiel parrot, pigeon) birds

PLOS ONE

Dear Dr. Damaziak,

Thank you for submitting your manuscript to PLOS ONE. After careful consideration, we feel that it has merit but does not fully meet PLOS ONE’s publication criteria as it currently stands. Therefore, we invite you to submit a revised version of the manuscript that addresses the points raised during the review process.

Please revise the manuscript by eliminating repetitions in the text as suggested by the reviewer.

We would appreciate receiving your revised manuscript by Jan 20, 2020. To enhance the reproducibility of your results, we recommend that if applicable you deposit your laboratory protocols in protocols.io, where a protocol can be assigned its own identifier (DOI) such that it can be cited independently in the future. For instructions see: http://journals.plos.org/plosone/s/submission-guidelines#loc-laboratory-protocols

We look forward to receiving your revised manuscript.

Kind regards,

Xiuchun Tian

Academic Editor

PLOS ONE

Reviewers' comments:

Reviewer's Responses to Questions

**Comments to the Author**

1. If the authors have adequately addressed your comments raised in a previous round of review and you feel that this manuscript is now acceptable for publication, you may indicate that here to bypass the “Comments to the Author” section, enter your conflict of interest statement in the “Confidential to Editor” section, and submit your "Accept" recommendation.

Reviewer #1: All comments have been addressed

2. Is the manuscript technically sound, and do the data support the conclusions?

Reviewer #1: Yes

3. Has the statistical analysis been performed appropriately and rigorously? 

Reviewer #1: I Don't Know

4. Have the authors made all data underlying the findings in their manuscript fully available?

Reviewer #1: Yes

5. Is the manuscript presented in an intelligible fashion and written in standard English?

Reviewer #1: No

6. Review Comments to the Author

Reviewer #1: I strongly recommend the authors to polish the draft more and more. For example, there are quite similar contents, "... is primarily composed of glycoproteins of the zona pellucida", "The components of IL (mainly glycoproteins of the zona pellucida)" and "The IL primarily consists of glycoproteins, five of which have been identified..." in lines 52, 55 and 60, respectively. For other example, the order of bird species in lines 39 to 40 do not depend on that in lines 36 to 37. The authors should arrange context to reduce such unnecessary repeats and disorders to make the arguments clear.

7. PLOS authors have the option to publish the peer review history of their article (what does this mean?). If published, this will include your full peer review and any attached files.

Reviewer #1: No

---

## [Author Response · Author response to Decision Letter 2]

1 Jan 2020

Response to the Reviewers 

Manuscript: PONE-D-19-16659R2

Dear Editor,

We would like to express our gratitude for assessing our manuscript entitled “Characterization of structure and protein of vitelline membranes of precocial (ring-necked pheasant, gray partridge) and superaltricial (cockatiel parrot, domestic pigeon) birds.” We conducted the research and prepared the manuscript with the greatest possible precision and care. We have stated the corrections for the specific remarks below. All the amendments have been made to the text and highlighted in yellow. Due to changes in the content of the text, the line numbers may not match those in the unreviewed manuscript.

R: I strongly recommend the authors to polish the draft more and more. For example, there are quite similar contents, "... is primarily composed of glycoproteins of the zona pellucida", "The components of IL (mainly glycoproteins of the zona pellucida)" and "The IL primarily consists of glycoproteins, five of which have been identified..." in lines 52, 55 and 60, respectively. For other example, the order of bird species in lines 39 to 40 do not depend on that in lines 36 to 37. The authors should arrange context to reduce such unnecessary repeats and disorders to make the arguments clear.

A: We have carefully checked the entire manuscript. All similar contents has been removed. The names of the birds have also been standardized and are now presented in the same order wherever possible.

---

## [Editor Report · Decision Letter 3]

14 Jan 2020

Characterization of structure and protein of vitelline membranes of precocial (ring-necked pheasant, gray partridge) and superaltricial (cockatiel parrot, domestic pigeon) birds

PONE-D-19-16659R3

Dear Dr. Damaziak,

We are pleased to inform you that your manuscript has been judged scientifically suitable for publication and will be formally accepted for publication once it complies with all outstanding technical requirements.

With kind regards,

Xiuchun Tian

Academic Editor

PLOS ONE
---

## [Editor Report · Acceptance letter]

17 Jan 2020

PONE-D-19-16659R3 

Characterization of structure and protein of vitelline membranes of precocial (ring-necked pheasant, gray partridge) and superaltricial (cockatiel parrot, domestic pigeon) birds 

Dear Dr. Damaziak:

I am pleased to inform you that your manuscript has been deemed suitable for publication in PLOS ONE. Congratulations! Your manuscript is now with our production department. 

With kind regards,

on behalf of

Dr. Xiuchun Tian 

Academic Editor

PLOS ONE